# Chitooligosaccharide and Its Derivatives: Potential Candidates as Food Additives and Bioactive Components

**DOI:** 10.3390/foods12203854

**Published:** 2023-10-20

**Authors:** Ajay Mittal, Avtar Singh, Jirayu Buatong, Jirakrit Saetang, Soottawat Benjakul

**Affiliations:** 1International Center of Excellence in Seafood Science and Innovation, Faculty of Agro-Industry, Prince of Songkla University, Hat Yai 90110, Songkhla, Thailand; ajy.mittal@yahoo.com (A.M.); avtar.s@psu.ac.th (A.S.); jirayu.b@psu.ac.th (J.B.); jirakrit.s@psu.ac.th (J.S.); 2Department of Food and Nutrition, Kyung Hee University, Seoul 02447, Republic of Korea

**Keywords:** chitooligosaccharide, derivatives, foods, additives, nutraceutical, functional ingredient, applications

## Abstract

Chitooligosaccharide (CHOS), a depolymerized chitosan, can be prepared via physical, chemical, and enzymatic hydrolysis, or a combination of these techniques. The superior properties of CHOS have attracted attention as alternative additives or bioactive compounds for various food and biomedical applications. To increase the bioactivities of a CHOS, its derivatives have been prepared via different methods and were characterized using various analytical methods including FTIR and NMR spectroscopy. CHOS derivatives such as carboxylated CHOS, quaternized CHOS, and others showed their potential as potent anti-inflammatory, anti-obesity, neuroprotective, and anti-cancer agents, which could further be used for human health benefits. Moreover, enhanced antibacterial and antioxidant bioactivities, especially for a CHOS-polyphenol conjugate, could play a profound role in shelf-life extension and the safety assurance of perishable foods via the inhibition of spoilage microorganisms and pathogens and lipid oxidation. Also, the effectiveness of CHOS derivatives for shelf-life extension can be augmented when used in combination with other preservative technologies. Therefore, this review provides an overview of the production of a CHOS and its derivatives, as well as their potential applications in food as either additives or nutraceuticals. Furthermore, it revisits recent advancements in translational research and in vivo studies on CHOS and its derivatives in the medical-related field.

## 1. Introduction

Annually, crustacean processing wastes are generated at 6–8 metric tons (MT) across the globe, and approximately 1.5 MT was reported in Southeast Asia [1]. The biowaste generated during shrimp processing is unavoidable, and disposal has become a major problem for processing industries along with the increased expenditure. It can produce an adverse effect on the environment (i.e., pollution), which creates risks to human health when the disposal management of waste is carried out improperly. Nevertheless, these biowastes contain several valuable compounds when appropriate processing or technology is implemented to earn better profit via their valorization. Shrimp processing waste contains a wide range of valuable bioactive compounds such as astaxanthin, chitin, bioactive peptides, fatty acids, amino acids, etc. [2]. Also, shrimp shell is cardinal waste from shrimp processing industries. It contains a high amount of chitin, which can be converted to its derivatives, especially chitosan (CS) and CS can be further converted to chitooligosaccharide (CHOS) [3].

CHOS (β-1-4-linked d-glucosamine) is a water-soluble depolymerized product of CS extracted from shrimp shells and other crustacean wastes [4,5,6]. It is positively charged cationic oligosaccharide with an average molecular weight (MW) and degree of polymerization (DP) of less than 3.9 kDa and 20, respectively. Sometimes, much larger molecules (up to 20 kDa) are also called CHOS. Their superior characteristics, such as a low molecular weight, low polymerization degree, and high-water solubility, compared to those of chitin and CS make them more applicable. Various methods including physical, chemical, and enzymatic methods have been employed for CS hydrolysis by cleaving glycosidic bonds to produce CHOS [7]. In terms of their applications, CHOS has been widely used as antioxidant, antibacterial, and antifungal agents [8,9] and vectors in gene therapy [10]. The physical and biological activities of CHOS are governed primarily by the DP, MW, and DD.

CHOS possesses the reactive amino group at C-2 and hydroxyl groups at C-3 and C-6, which could favor chemical modification such as acylation, oxidation, etherification, graft copolymerization, cationization, phosphorylation, etc. [11], thus enhancing CHOS’ bioactivities. CHOS derivatives have been used to alleviate metastasis and tumor growth [12,13] and increase bone strength by preventing osteoporosis [14,15,16]. It also showed immunomodulatory effects [17] and alleviated serum glucose levels in patients suffering from diabetes [18], indicating health promotion or disease prevention properties. Recently, efforts have been intensified to conjugate polyphenols (PPNs) with CHOS to improve their physicochemical and biological properties. Furthermore, CHOS derivatives have been incorporated into food and food products for quality improvement and shelf-life extension. Therefore, the objective of this review article is to provide updated information regarding the preparation and characterization of CHOS and their derivatives. Furthermore, the bioactivities and applications of CHOS and CHOS derivatives in food as well as its nutraceutical aspects are also revisited.

## 2. Preparation and Characterization of CHOS

A CHOS is a hydrolyzed product of CS, which has been extracted from crustaceans, fungi, etc. [19]. In general, the linkage of several glucosamine units via glycosidic bonds resulted in the formation of CS. A glycosidic bond is unstable and easy to be cleaved via various hydrolyzing agents including acids, enzymes, etc., in which CHOS or CS oligomers with variousDP can be produced [7,9,20]. In general, the DP of a CHOS ranges between 2 and 20 [20,21]. The hydrolysis of CS is performed via several methods with varying advantages and disadvantages.

### 2.1. Chemical or Non-Enzymatic Methods

#### 2.1.1. Acid Hydrolysis

CHOS can be produced utilizing mineral acids like hydrochloric acid, or a mixture of acids and electrolytes, e.g., nitrous acid, phosphoric acid, and hydrofluoric acid, as well as oxidizing agents like hydrogen peroxide or hydrogen persulfate [22]. In addition, acetic, trichloroacetic, lactic, and formic acids were also used for hydrolysis [21,22]. Several investigations have been conducted to better understand the process of CS hydrolysis via acids (Table 1) [23,24,25]. In acid hydrolysis, the protonation of oxygen at the glycosidic linkage (β-1,4-linkage) is started, followed by the addition of water to the reducing sugar end. Finally, the decomposition of protonated glycosidic linkages takes place [23,26]. Also, a two-step hydrolysis of CS was proposed [23]. These include (i) glycosidic linkage protonation and (ii) the cleavage of macromolecular chains into two smaller fragments (Figure 1A). Thus, the reaction begins with the attachment of a proton (H_3_O^+^) to the glycoside bond, and the subsequent scission of macromolecules into smaller ones is carried out [23]. Several factors affecting CS hydrolysis have been studied. The hydrolysis rate is mainly governed by the MW and DDA of CS [21,27]. Einbu, Grasdalen, and Vårum [25] also suggested that the hydration of the glycosidic oxygen atom resulted in the formation of a conjugate. A further breakdown of the exocyclic O-5 to C-1 bond generates a cyclic carbonium–oxonium ion. When the hydrolysis starts, the rate of hydrolysis is also affected by the presence of a glycosidic bond. In comparison to the other glycosidic links in chitin and CS chains, the glycosidic bond near the terminal residues is hydrolyzed at rates that are 2.5 and 2.0 times faster, respectively [25]. Also, the glycosidic bond located close to the non-reducing end is hydrolyzed at the fastest rate. Additionally, the glycosidic bond hydrolysis rate constants in GlcNAc_4_ are 50 times higher than those in GlcN_4_, which is connected to the *N*-acetyl group’s catalytic activity and the presence of the positively charged amino group on *N*-deacetylated sugar residue [25]. The acid hydrolysis of CS has been widely investigated, which provides different hydrolysis mechanisms. However, under harsh conditions, DP is difficult to be controlled. In addition, some secondary products can be formed, which are toxic [20]. Hence, the use of acid in CS hydrolysis is limited.

#### 2.1.2. Oxidative Hydrolysis

Since acids have certain limitations, chemical processes like oxidative hydrolysis have been routinely used to create water-soluble CS oligomers or CHOS [28,29,30]. Oxidizing agents such as hydrogen peroxide (H_2_O_2_) are known as powerful oxidizing agents, which could form reactive hydroxyl radicals and break the β-1-4 glycosidic linkages to produce CS oligosaccharide or CHOS [30]. Several reports are available on the hydrolysis of CS with the aid of H_2_O_2_ (Table 1) [30,31,32]. Trong, Le Nghiem, Phuoc, Du Bui, and Nguyen [32] prepared water-soluble CS via hydrolysis (MW: 90 kDa) using 2% H_2_O_2_ through heterogeneous and homogeneous hydrolysis. The obtained CHOS (~2000 g/mol) was water-soluble at a 10% (*w*/*v*) concentration. Similarly, Mittal, Singh, Hong, and Benjakul [30] prepared CHOS having DP of 3–6 and a MW of 1.2 kDa from shrimp shell CS (MW: 2 × 10^4^ kDa). Although H_2_O_2_ alone had the potential to hydrolyze CS, hydroxyl radicals were generated at a low content. Therefore, different catalysts such as ascorbic acid, processing technology (e.g., γ-radiolysis), ozone, and a transition metal ion (such as Fe, Cu, etc.) along with H_2_O_2_ have been used to enhance the generation of hydroxyl radicals [30,33]. Xia, Wu, and Chen [31] prepared a CHOS (DP: 2–9) using H_2_O_2_ in the presence of phosphotungstic acid in a homogeneous phase with a yield of 92.3%. Phosphotungstic acid is a thermally stable and acid-resistive heteropoly acid, which can be prepared simply. It has a high reactivity and is non-corrosive [34]. Huang et al. [35] documented a 99.32% increase in the hydrolysis of CS at 70 °C for 20 min in the presence of 0.04 g phosphotungstic acid compared to the control (without phosphotungstic acid), which only had 43% hydrolysis. Thus, degradation was inefficient when H_2_O_2_ was used alone. Moreover, when the concentration of phosphotungstic acid was increased from 0.04 to 0.1%, the rate of formation of reducing sugar was increased exponentially. Recently, Mittal, Singh, Hong, and Benjakul [30] used the redox pair method (Figure 1B), in which 0.05 M of ascorbic acid and 0.1 M of H_2_O_2_ yielded CHOS with the lowest DP (2–8) and MW (0.7 kDa) compared to the use of H_2_O_2_ alone. The DP and MW of the CHOS were also influenced by the hydrolysis time, temperature, concentration of reagents, etc., -as documented by Gonçalves, Ferreira, and Lourenço [34].

### 2.2. Physical Methods

CHOS has been prepared using physical methods such as microwave, lambda radiation, and high-intensity ultrasonication [34]. The microwave method mainly involves two steps. The first step is related to the shear process, which causes the oscillation of molecules and induces their breakdown, and the second step is thermal degradation [36]. Wasikiewicz and Yeates [36] reported that microwave radiation alone reduces the MW of CS to 30 kDa in an aqueous acetic acid solution without the use of other oxidative reagents (Table 1). The physical method consumes high energy, but it provides a lower DDA for the CHOS [22]. As a result, the physical approaches for the depolymerization of CS have not been widely used. Nevertheless, these hydrolysis methods can be used as pretreatments of CS before chemical or enzymatic hydrolysis [37].

### 2.3. Enzymatic Hydrolysis

The most common glycosyl hydrolase used to produce CHOS via the hydrolysis of the β-1,4-glycosidic bond of CS is chitosanase (EC 3.2.1.132) [38,39]. The use of chitosanase to produce CHOS, especially via the fermentation process, has been reported [40,41,42]. However, the scarcity and high cost of chitosanase limit its commercial application for the synthesis of CHOS. As a result, non-specific enzymes, including lipase, carbohydrase, proteases, etc., have been employed for CS hydrolysis [20,43]. Renuka et al. [44] used proteolytic enzymes (papain and pepsin) and carbohydrases (α-amylase and β-amylase) to generate a CHOS from shrimp shell CS (DDA: 76.43 and MW: 111.18 kDa). The obtained CHOSs had the MW range of 5.1–7.45 kDa as determined using the viscometric method. Among those enzymes, pepsin yielded CHOS with the lowest MW. Similarly, Roncal et al. [45] reported that the CHOS produced by pepsin showed a lower DP (16), which also indicated a lower MW after 20 h at 40 °C and pH 4 compared to those prepared using cellulase and lipase A (Table 1). This was likely attributed to the lower pH of the CS solution, in which pepsin exhibited the highest activity [39]. Lee et al. [46], Rokhati et al. [47], and Gohi et al. [48] also used lipase, pepsin, and carbohydrases such α- and β-amylase. A CHOS was successfully generated via CS hydrolysis. The pH of the CS solution was typically between 4 and 5, except for pepsin, whose pH range was between 2 and 3.5, thus favoring pepsin activity in an acidic environment [46,47,48]. Normally, polysaccharides are mainly hydrolyzed at a glycosidic bond. CS can be hydrolyzed at four different linkages in CS (-GlcN-GlcN-, -GlcN-GlcNAc-, -GlcNAc-GlcN-, and -GlcNAc-GlcNAc-) [9,39,45,46]. When squid pen CS was hydrolyzed with three distinct enzymes (lipase, amylase, and pepsin) at pH 5 for the same hydrolysis time (72 h), the obtained CHOS had the varied DDP [9]. Therefore, various enzymes can produce CHOS having different DDPs and MWs. In addition, the pretreatment of CS before enzyme hydrolysis can provide more cleavage site accessibility to the enzymes. Qian et al. [49] pretreated CS with H_2_O_2_ before hydrolysis using α-amylase. The produced CHOS had a DP of 5.4 when 20 U/g of the enzyme was used for 1.5 h.

**Table 1 foods-12-03854-t001:** Preparation and characterization of chitooligosaccharides (CHOSs) prepared using various methods.

Methods	Hydrolysis Conditions	Analytical Techniques	Characteristics of CHOS	References
**Chemical**				
Hydrochloric acid	CS: 1% (*w*/*v*) HCl: 0.5 M Temperature: 65 °C; time: 36 h	Size exclusion chromatography; viscometry	MW: 73.8 kDa	[23]
	CS: 2 mg HCl: 12.07 M Temperature: 40 °C; time: 28 h	Size exclusion chromatography	DP: 1	[24]
	CS: 10 mg/mL HCl: 0.1 M Temperature: 83 °C; time: 30 h	^1^H NMR; viscometry	DP: 13	[25]
Hydrogen peroxide	CS: 100 mL at 1% (*w*/*v*) H_2_O_2_: 1 M Temperature: 60 °C; time: 2 h	^1^H NMR; ^13^C-NMR; gel permeation chromatography; FTIR; MALDI-TOF	MW: 1.2 kDa; DP: 3–13	[30]
	CS: 1000 mL at 1% (*w*/*v*) H_2_O_2_: 20 mL (30%, *v*/*v*) Phosphotungstic acid: 1 g Temperature: 65 °C; time: 40 min	FTIR	DP: 2–9	[31]
	CS: 10 g H_2_O_2_: 100 mL (2%, *v*/*v*) Phosphotungstic acid: 1 g Temperature: 25–28 °C; time: 24 days	^1^H NMR; gel permeation chromatography; XRD; FTIR	MW: 2.04 kDa	[32]
	CS: 1000 mL at 1% (*w*/*v*) H_2_O_2_: 3 mL (30%, *v*/*v*) Phosphotungstic acid: 0.04 g H_2_O: 17 mL Temperature: 70 °C; time: 20 min	FTIR; X-ray diffraction	MW: 4.7 kDa	[35]
Redox pair	CS: 100 mL at 1% (*w*/*v*) Ascorbic acid: 0.05 M H_2_O_2_: 0.1 M Temperature: 60 °C; time: 2 h	^1^H NMR; ^13^C-NMR; gel permeation chromatography; FTIR; MALDI-TOF	MW: 0.7 kDa; DP: 2–8	[30]
**Physical**				
Microwave	Power: 100 W; frequency: 2.46 GHz; time: 20 min	Static light scattering; gel permeation chromatography; FTIR; ^1^H-NMR	MW: 30 kDa; DDA: 91% via ^1^H-NMR	[36]
**Enzymatic**				
Lipase	E: 8% (*w*/*w*) of CS Temperature: 50 °C; pH: 5.0; time: 12 h	Viscometry	MW: 79 kDa	[9]
Pepsin	E/S: 1:100 (*w*/*w*) of CS; temperature: 50 °C; pH: 4.5; time: 16 h	Viscometry, FTIR, differential scanning calorimetry	DDA: 84%, M_v_: 5.1 kDa	[44]
	E/S: 1:100 (*w*/*w*) of CS; temperature: 40 °C; pH: 4.5; time: 1 h	Viscometry, HPLC	DP: 16	[45]
Chitosanase	E: 1700 U/mg CS Temperature: 40 °C; pH: 5.0; time: 24 h	Gel permeation HPLC	DP: 3–7	[41]

The structural alternation of CS associated with CHOS production has been monitored via several methodologies including viscometry, chromatography, ^1^H-nuclear magnetic resonance (NMR), ^13^C-NMR, Fourier transmission infrared spectroscopy (FTIR), matrix-assisted laser desorption/ionization time-of-flight (MALDI-TOF), mass spectroscopy (MS), etc. [50]. After CS hydrolysis, a reduction in the MW of CS can be determined using the intrinsic viscosity and Mark–Houwink equation with the help of the Ubbelohde capillary viscometer (∅ = 0.5 mm) at 25 °C. Singh, Benjakul, and Prodpran [9] observed reductions in the intrinsic viscosity of squid pen CS from 3.79 to 0.41 dL/g and the average MW from 150 to 79 kDa during CHOS production. In addition, Mittal, Singh, Hong, and Benjakul [30] also determined the MW of a shrimp shell CHOS (0.7 kDa) using gel permeation chromatography. The structure of the CHOS was analyzed using ^1^H-NMR and ^13^C-NMR. In the ^1^H-NMR spectrum of CHOS, the signals at 4.92, 3.79–3.60, 3.06, 2.09, and 1.94 ppm were attributed to H1 (GlcN), H3-H6 (pyranose ring), H2 (GlcN), protons at C6, and acetyl protons, respectively [51]. The ^13^C-NMR of the CHOS had characteristic peaks at 21.84, 55.68, 59.90, 69.99, 77.21, 74.77, 97.50, and 179.15 ppm for carbon numbers 8, 2, 6, 5, 3, 4, 1, and 7, respectively [51]. MALDI-TOF has been used to study the distribution or diversification of oligomers in CHOSs. In addition, a sequence analysis of a hetero CHOS was carried out using the methods of derivatization and MALDI-TOF post-source decay (PSD) MS [50].

## 3. Preparation of CHOS Derivatives

### 3.1. Carboxylated CHOS

Carboxylated CHOS (C-CHOS) was prepared by grafting the carboxyl group (COCH_2_CH_2_COO-) to the amino site at C-2 of the CHOS [52]. The CHOS (6.5 g) was solubilized in 10% acetic acid (50 mL) and mixed with fifteen milliliters of methanol. Later, to prepare C-CHOS with various levels of substitution, succinic anhydride (6.6 g) was dissolved in acetone and added to the aforementioned mixture for 1 h at room temperature. At a pH of 9.0–10.0, the reaction mixture was agitated for 4 h. The pH was maintained using sodium carbonate. The degree of substitution of the carboxyl group per CHOS monomer was 0.90 [52]. However, DS and site specificity including the -NH_2_ and -OH groups denoted as *N* and *O* at C2 and C6, respectively, are affected by reaction conditions [53]. Therefore, the laccase/2, 2, 6, 6-tetramethylpiperidine-1-oxyl (TEMPO) oxidation system was applied, and only the hydroxyl group at C-6 of the CHOS could be oxidized into carboxyl groups. 6-carboxylate chitooligosaccharide (6-CCHOS) was prepared using the aforementioned method, in which a carboxylate ion content of 2.3 mmol/g 6-CCHOS was obtained [54].

The synthesis of the C-CHOS was confirmed using NMR (^1^H-NMR and ^13^C-NMR) [52]. ^1^H-NMR showed protons associated with -CH_3_, C1–6, and -CH_2_CH_2_- at 1.9, 2.6, 3.3–3.6, 4.5, and 2.4 ppm, respectively. ^13^C-NMR showed carbon related to N-CH_3_, -CH_2_CH_2_- close to the carboxyl group, C-2, C-6, C-3, C-4, C-5, C-1, amide C=O, and C=O at 22, 32, 56, 61, 69, 73, 76, 78, 102, 174, 176, and 180 ppm, respectively. Moreover, FTIR spectra showed peaks associated with hydroxyl groups at 3408 cm^−1^; alkyl stretching at 2931 and 2860 cm^−1^; ester C=O at 1723 cm^−1^; amide C=O at 1653 cm^−1^; carboxyl C=O at 1565 and 1408 cm^−1^, and a pyranose ring at 1112, 1068, and 1030 cm^−1^. Additionally, the derivatization of the CHOS to 6-CCHOS was confirmed via FTIR, in which a higher absorption intensity at 3363 cm^−1^ was found compared to the native CHOS. The band was observed at 1643 cm^−1^. This was due to carboxylate (C=O) and the absence of the amide I band [54]. Moreover, the peak associated with -COO appeared at 170 ppm when the ^13^C-NMR spectra were analyzed. These modifications led to the formation of hydrogen bonds connecting with hydrophilic groups.

### 3.2. Amino-Derived CHOS

Amino-derived CHOSs, namely aminoethyl CHOS (AE-CHOS), dimethylaminoethyl CHOS (DMAE-CHOS), or diethylaminoethyl CHOS (DEAE-CHOS), were prepared using 2-chlorethylamino hydrochloride, 2-dimethylamino-ethylchloride hydrochloride, or 2-diethylamino-ethylchloride hydrochloride, respectively [55,56,57,58]. The CHOS (0.40 g) was combined with the aforesaid amino solution (3.0 M; 20 mL) and mixed at 40 °C. After stirring the reaction mixture for 48 h, 3.0 M NaOH (20 mL) was added dropwise and filtered. The reaction mixture was then dialyzed against water after being acidified with 0.1 N of HCl. After the derivatization of the CHOS, the hydroxyl group at the C-6 position was replaced by the amino group due to its highest reactivity for aminoethylation [55].

In the FTIR spectrum of the synthesized CHOS amino derivatives, the peaks of absorptions at 2965 cm^−1^ and 1000–1150 cm^−1^ due to C–H stretching and C–O–C stretching were observed, respectively, indicating the substitution of the hydroxyl group at C-6 of the CHOS by AE, DMAE, and DEAE for AE-CHOS, DMAE-CHOS, and DEAE-CHOS, respectively. When an amino-derived CHOS was confirmed via ^1^H NMR, a peak appeared at 2.8 ppm, corresponding to the proton of –CH_2_N; a peak for protons of the acetyl group of the CHOS appeared at around 2 ppm; and peaks appeared at 2.9–3.6 ppm for protons of the pyranose unit superimposed with -NH_2_ of the aminoethyl group.

### 3.3. Sulfated CHOS

Sulfated CHOS (S-CHOS) was prepared by adding sulfonated groups to C-3 and C-6 of CHOS [59]. First, solution I was prepared by mixing the CHOS (1.5 g), dimethylformamide (DMF; 60 mL), and dichloroacetic acid for 24 h. To make a sulfonated reagent (solution II), chlorosulfonic acid (10 mL) was gradually added to DMF (60 mL) at 0–4 °C. After reaching 60 °C, solution II was progressively combined with solution I. The reaction was run for 2 h. Subsequently, 100 mL of distilled water was added, and NaOH (20%, *w*/*v*) was used to change the pH to 7.0. After centrifugation, the supernatant was collected and precipitated with ethanol. After that, water was added to the precipitate, and it was dialyzed. S-CHOS powder was produced by lyophilizing the concentrated solution [59]. Wang et al. [60] also made S-CHOS. In brief, chlorosulfonic acid (3.4 mL) was poured dropwise into an ice-cold three-necked flask containing DMF (17 mL). To obtain a sulfation reagent, the reaction was performed at room temperature for 1.5 h. Then, 1 g of CHOS was added to the sulfation reagent and agitated for 1 h at 70 °C. Ethanol was used to precipitate the reaction and repeatedly rinsed. The precipitate was dissolved in distilled water and the pH was adjusted to 8.0 using NaHCO_3_ solution. The resultant solution was dialyzed and lyophilized to obtain S-CHOS [60].

NMR and FTIR were used to analyze the basic structure of the S-CHOS as well as to identify the sulfate groups in the S-CHOS. The FTIR spectra revealed bands characteristic of the S-CHOS at 1221 and 813 cm^−1^, which were attributable to the S=O and C-O-S bonds, respectively. The hydroxyl groups were therefore successfully sulfated. After sulfation, the signals of C-6 in the ^13^C NMR spectra of the S-CHOS were displaced to 66.8 ppm from 59.9 ppm in the CHOS. Furthermore, the signal at 60.0 ppm for unsubstituted C-6 demonstrated partial sulfation. The sulfate substitution in the S-CHOS was 1.02, implying that both C-2 and C-3 were also largely sulfated throughout the process.

### 3.4. Quaternized CHOS

Quaternized CHOS (Q-CHOS) was synthesized via the conjugation of quaternary ammonium salt such as glycidyltrimethylammonium chloride (GTMAC) to the amino group at C-2 of CHOS under neutral and alkaline conditions. The hydroxyl group could react with the epoxide ring. Consequently, quaternization was carried out in an acidic environment. Feng et al. [61] prepared a Q-CHOS as follows: a CHOS (1 g) was dissolved in distilled water (20 mL) and mixed with three molar equivalents of GTMAC. Thereafter, the pH of the mixture was adjusted to 5.5 using acetic acid and heated at 50 °C for 12 h, and then the solution was precipitated using acetone followed by filtration and drying at 50 °C for 24 h. Also, Kim et al. [62] prepared a Q-CHOS using a similar method.

The FTIR spectra of the Q-CHOS showed characteristic bands as follows [62]: The amino group’s interaction with the GTMAC epoxide had an absorption peak at 1516 cm^−1^, showing a lower intensity, and a new peak caused by the methyl groups of GTMAC was discovered at 1479 cm^−1^. All of the pyranose ring protons for the Q-CHOS’s ^1^H-NMR were found between 3.3 and 4.6 ppm. At 3.4 and 2.5 ppm, proton signals from two types of methylene groups were detected, respectively. Moreover, a peak at 4.3 ppm was attributed to methine protons. In addition, protons of the trimethylammonium group appeared at a peak of 3.2 ppm in the Q-CHOS spectra.

### 3.5. N-Aryl CHOS

The synthesis of the *N*-aryl CHOS (aryl-CHOS) included a two-step reaction, in which the first step involved the formation of a Schiff base as an intermediate product and its reduction to aryl-CHOS occurred in the second step. CHOS (1%, *w*/*v*) was dissolved in ethanol (69%, *v*/*v*), followed by an adjustment of the pH to 5.0 using acetic acid (1%, *v*/*v*). The Schiff base was then created by adding 4-hydroxybenzaldehyde to the mixture and stirring for 12 h at room temperature. The reaction mixture was then added with 0.1 g of NaBH_4_ and agitated for 12 to 14 h. NaOH (15%, *w*/*v*) was used to neutralize the reaction mixture and centrifuged to separate the precipitate. Using dialysis membranes, the precipitate was dialyzed against distilled water. To eliminate the free aldehyde, the dialysate was precipitated in acetone and rinsed with diethyl ether. The final step was to dry the product at room temperature to produce 4-hydroxybenzyl-CHOS [63].

According to the FTIR spectra of 4-hydroxybenzyl-CHOS, characteristic bands appeared at 3195 cm^−1^ and were assigned to the C-H stretching of a benzene ring, whereas bands at 1506 and 1462 cm^−1^ indicated C-C bonds of a benzene ring. From the ^1^H-NMR spectra, 4-hydroxybenzyl-CHOS showed peaks at 8–6.9 ppm belonging to the protons of the aromatic ring of 4-hydroxybenzaldehyde along with characteristic peaks of CHOS. Thus, the spectroscopic results elucidated the successful link between 4-hydroxybenzaldehyde and CHOS [63].

### 3.6. Polyphenol (PPN)- or Phenolic Acid (PA)-Conjugated CHOS

Currently, the interest in PPNs or PAs has been rising because of their ubiquitous nature and health benefits. Moreover, CHOS was grafted with PPNs or PAs due to their excellent bioactivities such as their antioxidant, antimicrobial, antidiabetic, and anti-cancer properties [64]. In addition, enhanced bioactivities of CHOS were reported after conjugation with PPNs or PAs. CHOS-PPN or CHOS-PA conjugates were synthesized through a carbodiimide-based chemical coupling method and free radical grafting reaction [6]. In the carbodiimide-based chemical coupling method, carbodiimides such as 1-ethyl-3-(3-dimethylaminopropyl) carbodiimide (EDC) or *N*,*N*′-dicyclohexylcarbodiimide (DCC) are used. EDC or DCC can activate the carboxyl group to form an *O*-acylisourea intermediate, followed by coupling with the primary amine of a CHOS to yield amide bonds. Vo et al. [65] prepared a CHOS-gallic acid conjugate. Firstly, CHOS (MW: 3–5 kDa; 2.50 g) was dissolved in distilled water (20 mL) and methanol (40 mL). Then, the pH of the mixture was adjusted to 6.8 to produce so-called solution A. Simultaneously, gallic acid (0.94 g) was dissolved in methanol (10 mL) and mixed with a DCC-methanol mixture (1.0315 g DCC in 10 mL methanol), known as solution B. Both solutions (A and B) were gradually mixed at 30 °C for 5 h. Thereafter, the reaction mixture was filtered to remove dicyclohexyl urea, which was formed as a side product, and the remaining solution was kept at 2 °C overnight. Subsequently, the remaining solution was precipitated using diethyl ether, and the precipitate was dissolved in distilled water (20 mL), followed by dialysis to remove the free gallic acid. CHOS-gallic acid conjugate powder with a yield of 43.5% was obtained after the lyophilization of the dialysate [65]. Nevertheless, *O*-acylisourea is the intermediate product, which is prone to hydrolysis, and it regenerates into a carboxyl group [66]. It could lower the grafting efficacy. To conquer this, other coupling reagents such as *N*-hydroxysuccinimide (NHS) and 1-hydroxybenzotriazole (HOBt) were introduced to enhance the grafting efficiency and avoid side reactions. NHS or HOBt can convert unstable *O*-acylisourea to ester, which finally reacts with the amino and hydroxyl groups of CHOS to yield a CHOS-PPN or CHOS-PA conjugate. Eight different PAs including caffeic, ferulic, 4-hydroxybenzoic, protocatechuic, *p*-coumaric, syringic, sinapinic, and vanillic acids were conjugated with CHOS [67]. A CHOS (1 g) was dissolved in 100 mL of methanol (20%, *w*/*v*). Subsequently, the aforementioned PAs were mixed with one equivalent of dicyclohexylcarbodiimide (DCC) and 1-hydroxybenzotriazole (HOBt) each and three equivalents of triethylamine (TEA). The mixture was transferred into a CHOS solution and stirred at room temperature for 24 h. The resultant solution was precipitated and washed using acetone, followed by lyophilization to obtain a PA-conjugated CHOS. The degree of substitution was in the range of 5.7–10.3%, depending on the PA used for conjugation. Moreover, the grafting of PAs on the CHOS was confirmed with the aid of ^1^H-NMR, in which peaks associated with the aromatic proton signals were observed between 6.0 and 8.0 ppm. In addition, PA-acylated CHOS derivatives were also produced using the carbodiimide coupling method [68]. The reaction was governed via amide bonding using the EDC/NHS catalytic system. PAs, including gallic acid, ferulic acid, *p*-coumaric acid, caffeic acid, protocatechuic acid, sinapic acid, and salicylic acid (25 mM each), were dissolved separately in deionized water or anhydrous ethanol at pH 5.0. Subsequently, 50 mM of EDC and 50 mM of NHS were added into the aforementioned solution and mixed for 2 h, followed by the addition of a CHOS (10 mM). After being stirred for 24 h at room temperature, precipitation was carried out using ethanol. The PA-acylated CHOS was obtained via the lyophilization of the precipitate [68].

The carbodiimide-based chemical coupling method generally requires a large amount of chemical crosslinkers in the reaction. Moreover, these crosslinkers are expensive and have hazardous impacts on the environment. They may cause adverse impacts on humans. Thus, their usage in the food and pharma sectors is limited [6]. The free-radical-induced grafting method was discovered as an alternative and successful conjugation, and it has been widely applied for the grafting of PPNs with CHOSs (Figure 2). The chemicals utilized in this process are affordable, less harmful, and environmentally benign, which can increase CHOS applications in the biomedical and food industries. Therefore, it could be an effective alternative for CHOS grafting with PPNs. Ascorbic acid is subjected to oxidation for ascorbate ion formation, which further donates two hydrogen ions or electrons in the presence of H_2_O_2_. Ascorbate is ultimately transformed into the resonance-stabilized tricarbonyl ascorbate free radicals via ascorbate free radicals [69]. With a pKa of -0.86, these ascorbate free radicals exist as tricarbonyl ascorbate free radicals because they cannot be protonated [70]. The previously generated H-ion or electron then interacted with H_2_O_2_ to make hydroxyl radicals. Thereafter, to create CHOS-macroradicals, the hydroxyl radical was used to extract hydrogen from the functional groups (amino and hydroxyl) of the CHOS [70]. In the end, the PPN molecules formed covalent bonds with the CHOS macro-radicals, resulting in the production of the CHOS-PPN conjugate.

Recently, the free radical grafting technique has been utilized by Mittal, Singh, Zhang, Visessanguan, and Benjakul [64]. CHOS from shrimp shells was coupled with different polyphenols. At a pH of 5.0, the CHOS (1%, *w*/*v*) was dissolved in acetic acid (0.5%, *v*/*v*). CHOS solution was then added to 4 mL of H_2_O_2_ (1 M) containing 0.10 g ascorbic acid, which was incubated at 40 °C for 15 min to produce hydroxyl radicals. After this, the reaction mixture was left at room temperature for 1 h while being constantly stirred. Thereafter, gallic acid, ferulic acid, caffeic acid, catechin, and epigallocatechin gallate (EGCG) were added to the mixture to attain concentrations of 10–100% (*w*/*w* of CHOS). The mixture was then stored at an ambient temperature in the dark for another 24 h. The resultant mixtures were lyophilized to produce CHOS-PPN conjugate powders, which were subsequently dialyzed using a dialysis bag against distilled water. Overall, the concentrations and types of PPN or PA have impacts on the conjugation or grafting efficiency [71].

In another study, a squid pen CHOS was conjugated with EGCG through the free radical grafting method [69]. Ascorbic acid (0.2%) and 0.1 M of H_2_O_2_ were used to oxidize CHOS at varied concentrations (1 and 2%, *w*/*v*) for 2 h at 25 °C in the dark. The combinations were then agitated for another 24 h at 25 °C in the dark after the addition of solutions containing EGCG (0.1–1%, *w*/*v*). The resulting mixtures were lyophilized after being dialyzed for 48 h at 25 °C against distilled water.

CHOS was also conjugated with gallic acid and syringic acid [72,73,74]. CHOS (2%, *w*/*v*) was dissolved in distilled water, followed by the addition of 1 mL of H_2_O_2_ (1.0 M) containing 0.054 g of ascorbic acid. PA at different amounts was added to the mixture after 30 min of the preceding step. The reaction was performed at pH 5 with continuous stirring at room temperature for 6 h. Finally, the unreacted PA and catalyst were removed via filtration, and then they were lyophilized to obtain PA-CHOS powder.

By using an easy and safe method (ion exchange), Sun, Ji, Cui, Mi, Zhang, and Guo [68] were able to create PA-CHOS salts. In this process, protonated amino groups at C-2 of the CHOS bonded with carboxyl negative ions on PAs, forming ionic bonds with salicylic acid, gallic acid, protocatechuic acid, ferulic acid, caffeic acid, and p-coumaric acid. Depending on their solubility, the PAs (30 mM) were dissolved in either water or ethanol at 65 °C. Thereafter, the CHOS (10 mM) was solubilized in deionized water and added to a phenolic acid solution gradually. The reaction was performed at 65 °C for 24 h. The solution was precipitated and washed using ethanol, followed by lyophilization.

Yuan et al. [75] conjugated CHOS with caffeic acid through the *N*,*N*′-carbonyldiimidazole (CDI) catalysis method. Caffeic acid at various amounts/concentrations (0.90 g/5 mM; 1.80 g/10 mM; and 3.60 g/20 mM) and DMSO (5 mL) were added together. Then, CDI, equivalent to caffeic acid, was added and left for 30 min. Thereafter, the reaction mixture was heated at 60 °C for 12 h under the N_2_ atmosphere. This step concluded the imidazolide activation of caffeic acid, and it was added to a chitooligosaccharide lactate (3.22 g) dissolved in DMSO (10 mL). The mixture was maintained at 60 °C with continuous stirring for 12 h under N_2_. Finally, the reaction mixture was precipitated using acetone and washed using ethanol at room temperature, followed by lyophilization.

Synthesized CHOS-PPN or CHOS-PA conjugates were characterized using FTIR and NMR. Generally, the FTIR spectra showed (i) an absorption band around 1648 cm^−1^, indicating the formation of an amide linkage as a result of the conjugation between the –NH_2_ group of the CHOS and the -COOH group of PAs; (ii) C-C stretching vibrations and C-H out-of-plane bending vibrations due to benzene rings between 1450 and 1550 cm^−1^ and between 800 and 900 cm^−1^; (iii) stretching vibrations of the carbonyl group (C=O) at 1748 cm^−1^; and (iv) an ester bond between CHOSs and PAs at 1284 cm^−1^ [39,64]. Moreover, ^1^H-NMR confirmed the grafting of PPNs or PAs on CHOSs, in which proton signals corresponding to the CHOSs appeared along with aromatic hydrogen between 6.0 and 8.0 ppm. In addition, the CHOS-PPN or CHOS-PA conjugates had B and D aromatic rings between 100 and 170 ppm, while A and C aromatic rings were found between 58 and 80 ppm when analyzed using ^13^C-NMR [64].

### 3.7. Amphiphilic CHOS

In relation to their regulated rates of biodegradation and possible uses in drug administration, the production of amphiphilic biocompatible and biodegradable copolymers has attracted attention. Polycaprolactone (PCL) is a hydrophobic polymer that is extensively used as the implantable carrier for drug delivery systems. Thus, PCL was used to prepare an amphiphilic CHOS, which possessed self-assembly properties to encapsulate several drugs. Wang et al. [76] prepared a CHOS-PCL in two steps. Firstly, the functional groups (hydroxyl and amino groups) of the CHOS were selectively protected by hexamethyldisilazane (HMDS). Thereafter, a CHOS, either trisilylated or disilylated, at C-3 was dissolved in an organic solvent, such as chloroform or xylene or a mixture of both, with PCL and stannous octoate under N_2_. Then, the reaction mixture was kept at 120 °C for 24 h with continuous agitation. The resulting graft copolymer (CHOS-PCL) was then stirred at room temperature while being deprotected with the isopropyl alcohol/H_2_O/HCl solution. The maximum DS of 47.3% was determined using ^1^H NMR when PCL was grafted on the CHOS at different molar ratios. In addition, methylene proton signals of PCL were observed at 4.1, 2.3, 1.7, and 1.4 ppm along with proton signals associated with the methine and methylene of the CHOS at 3.0–5.0 ppm, thus confirming the grafting of PCL with the CHOS [76].

Some fatty acids such as arachidic acid (AA), stearic acid (SA), etc., were used for conjugation with CHOS to achieve amphiphilic properties. An amphiphilic CHOS-AA conjugate was synthesized [77]. Separately, at 50 °C for 15 min, CHOS (0.2 mM) and AA (0.6 mM) were dissolved in 20 mL of DMSO. Then, at room temperature, AA’s carboxyl groups were activated by adding EDC and NHS (1.5 mol/mol of AA). An activated AA solution was added dropwise to the CHOS solution within 5 min. Following a 12 h-incubation period for the coupling process, 4 mL of deionized distilled water was added to the reaction mixture. After applying 1 N of HCl to change the mixture’s pH to 3.5, it was agitated for an additional 30 min. Acetone was added to the mixture for precipitation. Free arachidic acid was then removed via centrifugation. The precipitate was dispersed with distilled water and dialyzed using a dialysis membrane against distilled water for 24 h. Finally, the dialyzed products were lyophilized [77].

The amine peak at 1591 cm^−1^ of the CHOS was replaced by a new absorption band at 1558 cm^−1^, which was connected to the development of a new amide bond (amide II band) based on the FTIR spectrum of the CHOS-AA conjugate. Additionally, the CHOS-AA conjugate’s absorption bands at 2918 and 2851 cm^−1^ indicated the stretching vibrations of the acyl chain and CH_2_ and CH_3_ of AA. Additionally, ^1^H-NMR indicated proton signals for both the CHOS and AA, suggesting the conjugation of AA to the CHOS. The conjugation was confirmed with the peak of CH_2_ associated with a carbonyl group of AA at 2.07 ppm, whereas that peak disappeared in the spectrum of a physical mixture of CHOS and AA [77].

In a similar manner, SA was grafted with a CHOS when EDC was present [78]. While stearic acid (0.5 g) was dissolved in 20 mL of ethanol, CHOS (0.4 g) was solubilized in 30 mL of distilled water. At room temperature, 2 g of EDC was added to the CHOS solution, which was then heated to 90 °C while being vigorously stirred. Parallelly, the SA solution was added dropwise. The final reaction mixture was stirred at 90 °C for 5 h, cooled to room temperature, and further stirred for 24 h. After the reaction was finished, the mixture was dried at 50 °C in a vacuum oven, and the residue was dissolved in 20 mL of ethanol. To eliminate the unreacted SA, the precipitate was filtered and then collected [78]. Finally, the precipitate was dissolved in distilled water, followed by dialysis and lyophilization, respectively. The peaks of amide I and II at 1640 and 1560 cm^−1^, respectively, were due to the amide band between the CHOS and SA, along with the absence of an absorption peak associated with the carboxyl groups of SA at 1700 cm^−1^ in the FTIR spectra of the CHOS-SA conjugate. This confirmed the successful grafting. Moreover, proton signals at 0.9 and 1.0 ppm due to the methyl and methylene hydrogen of the stearate group in the CHOS-SA also verified the grafting of SA on the CHOS [79]. In addition, the degree of substitution was 15.4%, as determined using the 2,4,6-trinitrobenzene sulfonic acid method. Hu et al. [80] further conjugated CHOS-SA with the antitumor drug, i.e., doxorubicin (DOX). The conjugation was confirmed via ^1^H-NMR, in which the proton of the anthracene of DOX at 8.0 ppm in DOX-CHOS-SA was recorded.

### 3.8. Other CHOS Derivatives

Guanidinylated CHOS was prepared by dissolving CHOS hydrochloride (1 g), 1-amidinopyrazole hydrochloride (2.70 g; 18.4 mmol), and trimethylamine (3.34 g; 33.0 mmol) in water for 7 days [81]. The mixture was precipitated using isopropanol and collected via filtration, followed by vacuum drying. Guanidinylated CHOS showed characteristic absorption peaks for the C=N stretching vibration and NH bending vibration of guanidino groups at approximately 1725 cm^−1^ and 1590 cm^−1^, respectively. Moreover, the ^13^C-NMR spectrum showed signals corresponding to quaternary carbons in the guanidino groups at 159.3 ppm [81].

## 4. Bioactivities of CHOS and CHOS Derivatives and Their Applications in Foods

### 4.1. Antioxidant Activity

Antioxidant activity indicates the ability of a compound to inhibit the oxidation of lipids, proteins, DNA, etc., by preventing the propagation of an oxidative chain reaction or the scavenging of a hydroxyl radical formed through Fenton’s reaction. These kinds of antioxidants are called primary antioxidants. Moreover, secondary antioxidants can chelate prooxidants such as Fe, Cu, etc. [82]. Generally, fatty foods are prone to the off-flavor and off-odor associated with lipid oxidation, especially polyunsaturated fatty acid (PUFA), thus resulting in a short shelf life [83]. To inhibit or control lipid oxidation, natural antioxidants such as CHOS, which can provide H-ion to fatty acid radicals with the help of a free amino group at the glucosamine unit, have been used [39,84,85]. In general, CHOS with higher MW and DP showed less antioxidant activity than those possessing lower MW and DP [86]. This could be because the free amino group of a CHOS can more readily scavenge free radicals and serve as a hydrogen donor. In comparison to the CS, the CHOS with various MWs in the range of 5–50 kDa showed increases in the DPPH radical scavenging activity, metal chelating activity, and reducing power [84]. In addition, the preparation method also affects the antioxidant activity. Mittal, Singh, Hong, and Benjakul [30] prepared a CHOS using oxidative hydrolysis via H_2_O_2_ in the absence and presence of ascorbic acid. CHOS produced using H_2_O_2_ and an ascorbic acid redox pair had a higher level of antioxidant activities compared to those produced using H_2_O_2_ alone (Table 2). This was more likely due to the lower MW and DP of the former CHOS. CHOS from squid was incorporated into tuna slices to inhibit the auto-oxidation of myoglobin, which could result in the formation of dark meat [87]. When 200–400 ppm CHOS was added to yellowfin tuna slices, a lower formation of metmyoglobin and color discoloration was noticed during 9 days of storage at 4 °C, irrespective of the modified atmospheric packaging (Table 2) [88]. The strong DPPH and ABTS scavenging activities, reducing power, and metal chelating activity of the CHOS contributed to fewer changes in the color [69]. This might help to prevent the autoxidation of myoglobin [69]. CHOS can reduce Fe^3+^ to Fe^2+^, which is associated with its FRAP. This could be related to the reduction in the formation of metmyoglobin [89]. Similarly, squid pen CHOS lowered the lipid oxidation, as indicated by the lower peroxide value and TBARS value of sardine surimi gel during 10 days of storage at 4 °C [9]. CHOS from squid pen also reduced the peroxide value, TBARS, and destruction of PUFA of Asian sea bass slices when treated with high-voltage cold atmospheric plasma (HV-CAP) [90]. In general, HV-CAP is known to produce reactive species, which can cause the lipid oxidation of fatty foods [90]. In another study, CHOSs improved the stability of emulsion prepared using shrimp oil, as indicated by the lower lipid oxidation than the emulsion added with α-tocopherol [91]. Moreover, CHOS prevented the degradation of carotenoids and PUFAs of emulsion to a higher extent [91].

Due to the excellent antioxidant activity of PPNs [92,93], their conjugation with CHOSs enhanced the antioxidant activities drastically (Table 3). Singh, Benjakul, Huda, Xu, and Wu [69] prepared a CHOS-EGCG conjugate and reported enhanced ABTS and DPPH radical scavenging activities due to the higher number of hydroxyl groups present in the structure. Similarly, CHOS was conjugated with different PPNs or PAs, namely gallic acid, catechin, EGCG, ferulic acid, and caffeic acid, in which the CHOS-catechin conjugate showed the highest ABTS, DPPH, and peroxyl radical scavenging activities along with metal chelation compared with the CHOS and the other CHOS-PPN conjugates [64]. In another study, CHOS conjugated with caffeic acid was able to inhibit 82% and 90% of DPPH and nitric oxide (NO) radicals, respectively [67]. It also had greater reducing power and hydroxyl radical scavenging activity than the other CHOS-PA conjugates. Generally, the conformation of a chelator, ionization potential, and the proton dissociation energy of antioxidants influence their antioxidant potential [94]. When gallic acid was conjugated with CHOS, the enhanced DPPH and ABTS radical scavenging and H_2_O_2_ scavenging activities were reported [95]. Furthermore, when a CHOS-caffeic acid conjugate with the highest grafting ratio was incorporated into a CS-based film, it was able to enhance the DPPH radical scavenging ability and reducing capacity without affecting the thermal stability of the film [75]. PA-CHOS salt derivatives prepared via the ion-exchange method had a higher antioxidant capacity than the PA-acylated CHOS derivatives synthesized using the EDC/NHS catalytic system [68]. This indicates that the preparation method directly affects the bioactivities of the resultant conjugates.

### 4.2. Antidiabetic Activity

Diabetes mellitus (DM) is the most common endocrine disorder caused by insulin production deficiency (T1DM) or a combination of insulin resistance action and insulin secretion from the pancreas (T2DM). Moreover, postprandial hyperglycemia (PH) and elevated blood sugar levels after meals could play crucial roles in the development of T2DM and prolonged PH, causing micro-vascular and macro-vascular complications [96]. T2DM patients generally had healthy β-cells for many years after the disease started, but a high glucose level could harm the cells [97]. Thus, it is essential to maintain tight control over β-cells and insulin secretion to enhance the health condition of T2DM patients. The antidiabetic potential of CHOS and its derivatives in both in vitro and in vivo models was reported (Table 2). α-amylase and α-glucosidase are two key enzymes involved in the metabolism of carbohydrates to mono units, which are absorbed into the blood via the intestinal tract and cause PH. The inhibitors of these two enzymes can slow down the breakdown of carbohydrates and lengthen the time it takes for carbohydrates to be digested. This decreases the pace at which glucose is absorbed. As a result, the postprandial rise in blood sugar is retarded [64]. CHOS (0.7 kDa) effectively reduce the activities of α-amylase and α-glucosidase by 20 and 19%, respectively [64]. Through mixed-type inhibition kinetics, CHOS reduced the activity of α-amylase. It could interact with the Asp197 and Glu233 amino acid residues found at the active site of α-amylase through hydrogen bonding [98]. CHOS plausibly interacted with amino acid residues (Arg 411, Asp 382, and Ala 59) of α-glucosidase associated with enzyme inhibition [64]. CHOS also showed potential to protect β-cells against high glucose by accelerating the proliferation of pancreatic β-cells, which resulted in a higher secretion of insulin to lower the glucose [99]. An in vivo study was conducted by Ju et al. [100] using STZ-induced T2D mice, in which glucose metabolism was improved via lowering the fasting blood glucose and insulin levels and upregulating the mRNA expression of GLUT4 in muscle and adipose tissue when the mice were fed with CHOS. Liu, Liu, Han, and Sun [18] further claimed that CHOS therapy might lower blood glucose levels, restore poor insulin sensitivity, and enhance the overall health and diabetic symptoms in diabetic rats. Moreover, the absorption of a CHOS in the body is strongly affected by the MW of CHOS, which further influences its antidiabetic potential. The hypoglycemic activities of CHOS with three different MWs (<1000 kDa, 1–10 kDa, and >10 kDa) in diabetic mice were also studied [101]. CHOS with the lowest MW efficiently controlled postprandial hyperglycemia by reducing the activity of hydrolyzing enzymes and enhancing the bloodstream uptake of glucose into the muscle and fat tissues. Later results were verified through the downregulation of intestinal SGLT1 and GLUT2 expression in Caco-2 cells and the decrease in α-glucosidase enzyme activity [102,103]. CHOS (DP: 4) effectively reduced the blood glucose and lipid levels in T2DM mice and improved insulin signaling through the downregulation of PEPCK, FBPase, and G6Pase mRNA gene expression in the IRS/AKT pathway along with the decrease in IRS-1 phosphorylation and an increase in AKT phosphorylation, which promote glycogen synthesis and inhibit gluconeogenesis [104].

CHOS derivatives, such as S-CHOS, with varying degrees of substitution protected pancreatic β-cells and MIN6 cells from H_2_O_2_-induced apoptosis in a dose-dependent manner (Table 3) [105]. Also, S-CHOS with a higher DS (1.9) effectively prevented apoptosis through the downregulation of H_2_O_2_-induced Bax mRNA expression, Caspase-3 mRNA expression, and NF-κB/p65 activation factor and the upregulation of Bcl-2 mRNA expression. Moreover, S-CHOS with a higher DS exhibited promising antioxidant properties for the treatment of oxidative diseases due to the critical role of the sulfate content [106]. In another study, CHOS-biguanide (CHOS-G) at 500 mg/kg BW was administered to STZ-induced diabetic rats through intragastric gavage for 8 weeks, and it could lower the blood glucose level by protecting the insulin signaling system, delaying β-cell apoptosis, improving β-cell function, and promoting insulin secretion. Therefore, using CHOS could be a novel strategy for preventing T2DM [107].

### 4.3. Anti-Inflammatory Activity

In inflammation, numerous factors, including physical injury, microbial invasion, UV radiation, and immunological responses, are involved. The etiology of several illnesses, including chronic asthma, rheumatoid arthritis, multiple sclerosis, inflammatory bowel disease, psoriasis, and cancer, can be aided by excessive or protracted inflammation (Vo et al., 2012). The anti-inflammatory property of CHOS was affected by several factors such as MW, DP, and DA [108]. It was proposed that a CHOS might function as an interleukin 6 (IL-6) and tumor necrosis factor α (TNF-α) induction-resistance protein 1 (AP-1) activator in macrophages (Table 2). The TNF-α, NO, and IL-6 levels in LPS-activated RAW264.7 macrophage cells could be decreased by CHOS at 500 g/mL [109]. Also, CHOSs can reduce reactive oxygen species formation, NF-κB upregulation, the phosphorylation of Erk1/2 and Akt, and Nrf2/HO-1 in LPS-exposed cells [109]. Three different types of CHOSs with an MW between 0.2 and 1.2 kDa, namely fully deacetylated (fdCOS), partially acetylated (paCOS), and fully acetylated (faCOS) CHOSs, were analyzed for their anti-inflammatory potential in LPS-induced RAW 264.7 cells [110]. The highest anti-inflammatory effect indicated by the alleviation in TNF-α production after 2 h of LPS simulation was obtained using 250 ng/well, irrespective of the CHOS type. However, fdCOS and faCOS were able to significantly reduce the production of TNF-α at 6 h after stimulation with LPS. Similarly, CHOS with 90% deacetylation showed a higher anti-inflammatory effect via the reduction in TNF-α production compared with a CHOS with 50% deacetylation [111]. Overall, the acetamido group of CHOSs could play a critical role in bioactivities. CHOS, administered at a maximum daily dose of 16 mg/kg MW, showed protective effects against lung inflammation induced by ovalbumin in mouse models of asthma. This protection was associated with a notable reduction in the levels of mRNA expression and proteins such as IL-4, IL-5, IL-13, and TNF-α in both lung tissue and bronchoalveolar lavage fluid [112]. Human lung epithelial A549 cells were employed to study the anti-inflammatory effects of CHOSs (MW: 3–5 kDa) and CHOSs conjugated with gallic acid at 50–200 μg/mL. Both forms of CHOS demonstrated the ability to reduce the production of inflammatory markers like TNF-α, IL-8, COX-2, and PGE2 in A549 cells stimulated with LPS in a dose-dependent manner. However, the CHOS-gallic acid conjugate had a more profound anti-inflammatory effect (Table 3) [65]. In Chang liver cells, 4-hydroxybenzyl-CHOS suppressed the expressions of inducible nitric oxide synthase (iNOS) and cyclooxygenase-2 (COX-2) to diminish NO and prostaglandin E2 (PGE2) synthesis [63]. Additionally, in a dose-dependent manner, 4-hydroxybenzyl-CHOS reduced the activation of mitogen-activated protein kinases (MAPKs), the translocation of nuclear factor-κβ (NF-κβ), and the degradation of inhibitory κβα (Iκβα). Thus, 4-hydroxybenzyl-CHOS may have anti-inflammatory properties (Table 3). Additionally, in LPS-induced RAW 264.7 cells, the apple polyphenol-CHOS microcapsule dramatically increased the cytokine IL-10 levels while decreasing the production of pro-inflammatory molecules, including NO and TNF-α, demonstrating the anti-inflammatory potential of CHOS [113].

### 4.4. Anti-Cancer Activity

In vitro and in vivo anti-cancer activities of CHOS and its derivatives have been documented (Table 2 and Table 3). The antiproliferative activity of CHOS was demonstrated in various types of cancer cell lines, including human lung cancer [114,115,116], hepatocellular carcinoma [12], gastric cancer [117], and colon cancer [118]. Moreover, various animal studies also provided supporting details that emphasized the anti-cancer activity of CHOS. With LM3 and HepG2 orthotopic liver cancer models, the anti-cancer mechanism of CHOS was revealed by upregulating the NF-κB pathway, which promoted the expression of the p53 protein, leading to the apoptosis of cancer cells [119]. The anti-cancer activity of CHOS was also found to be associated with immunomodulatory properties that attenuate the toxicity of chemotherapeutic drugs while enhancing the anti-tumor immune response. The combination of an anti-cancer drug, cyclophosphamide (CTX), and a CHOS showed the best tumor reduction outcome when sarcoma 180 (S180) residual-tumor mice were used as models [120,121]. This phenomenon was related to the increase in certain types of immune cells, including CD4+/CD8+ T lymphocyte, NK cell, tumor-infiltrating T cells, and macrophages [120,122].

The anti-cancer effect is also dependent on the characteristics of CHOS, including the MW, DP, DD, type of modification, and charge [121]. Park et al. [116] demonstrated that CHOSs with higher MWs showed less cytotoxic effects on prostate cancer cells, lung cancer cells, and hepatoma cancer cells than low-MW CHOS. The amount of high-MW CHOS used to inhibit 50% of cancer cell growth was two times higher than that of low-MW CHOS [116]. Another study also revealed that the levels of DD were responsible for anti-cancer activity, since a CHOS with a DD of 95% showed more efficiency in inhibiting the proliferation of cancer cell lines than a CHOS with a lower DD (90%) [115,116,123]. Moreover, the anti-cancer mechanism of CHOS has been postulated from the cationic charge at C-2 that facilitates the selective binding of CHOS on glycoproteins present on the cancer cell membrane [121]. This interaction consequently results in cell permeability loss [123]. Specifically, YKL-40 glycoprotein has been hypothesized to be a CHOS-targeted protein due to its overexpression on the cell membrane of various cancer cells [121]. This protein is related to cancer progression since it is associated with anti-apoptotic pathways and is also considered a pro-angiogenic factor [124,125].

In addition to the native structure, modified a CHOS has been documented to contain anti-cancer activity with improved efficiency (Table 3). Various kinds of water-soluble amino-derivatized CHOS with different substitution groups, including AE-CHOS, DMAE-CHOS, and DEAE-CHOS, were tested for their anti-cancer properties toward AGS gastric cancer cells [126]. AE-CHOS and DEAE-CHOS provided the highest abilities to inhibit cancer cell growth, which was owing to the regulations of p53, p21, Bcl-2, and Bax proteins [126]. When an AE-CHOS was prepared using a low-MW CHOS (<1 kDa), it demonstrated a significantly improved capability to inhibit cancer proliferation and promote cancer cell apoptosis [57]. Gallic acid (GA) is a PPN compound that was recently used to conjugate with a CHOS, and it displayed the ability to suppress the proliferation of cancer cell lines [64,127,128]. CHOS-GA could inhibit gastric cancer cell growth via the induction of the apoptosis pathway with coincidental upregulations of p53, p21, Bax, caspase 9, and caspase 3 proteins [128]. The proteomics study on CHOS-GA-treated SW620 colon cancer cell lines also revealed that the CHOS-GA increased the molecular pathways related to intermediate filament organization and structural constituents of the cytoskeleton, while the CHOS influenced the changes in the ribonucleoprotein complex, cytoplasmic translation, and nucleosome assembly [127]. In this study, keratin 18 was hypothesized to show the anti-cancer effects of the CHOS-GA [127,129]. Additionally, a carboxymethyl CHOS (CM-CHOS) was developed and evaluated for its anti-cancer activity in the H22 tumor-bearing mice model [130]. The treatment of CM-CHOS on tumor-bearing mice could suppress the growth of a tumor by promoting the phagocytosis and nitric oxide productions of mouse peritoneal macrophages and the induction of cancer cell apoptosis. The CM-CHOS had no toxicity to normal livers and other immune organs [130].

### 4.5. Antiviral Activity

During the past couple of years, coronavirus 2 (SARS-CoV-2) caused a pandemic around the globe. The antiviral capacity of CHOS was tested against SARS-CoV-2 (Table 2), and they showed the inhibition of SARS-CoV-2 through the suppression of both the RdRp and E genes [131]. Moreover, plaques were not developed by the SARS-CoV-2 virus when a CHOS at 25 µg/mL was used. In another study, Jang, Lee, Shin, Lee, Jung, and Ryoo [131] elucidated the effect of the MW of CHOS on SARS-CoV-2 inhibition. A CHOS with an MW of 30 kDa effectively reduced the expressions of the RdRp and E genes and resulted in less plaque formation compared to the other treatments. However, the exact mechanism of SARS-CoV-2 inhibition via CHOS is still unclear.

In addition, the modification of CHOS could enhance its antiviral efficacy (Table 3). S-CHOS was prepared to analyze their inhibitory potential toward influenza A virus (IAV) [60]. The S-CHOSs (IC_50_: 54.6 μg/mL) effectively inhibited the IAV effects in MDCK cells treated with IAV (MOI = 0.1) and showed a lower toxicity than CHOS (IC_50_: >3000 μg/mL). Moreover, the pretreatment of H5N1-Luc pseudovirus with S-CHOS (1000 and 500 μg/mL) significantly decreased the amount of infected H5N1 pseudovirus in MDCK cells. The oral administration of S-CHOS in mice significantly decreased the pulmonary viral titers and improved the survival rate in IAV-infected mice [60]. S-CHOS (MW: 3–5 kDa) demonstrated strong inhibitory effects against HIV-1 replication at non-toxic concentrations through the inhibition of HIV-1-induced syncytia formation, lytic effects, and p24 antigen production and blocked viral entry by disrupting the binding of HIV-1 gp120 to the CD4 cell surface receptor [132]. On the other hand, unsulfated CHOS had no activity against HIV-1, highlighting the importance of sulfation. Furthermore, the potential of CHOS conjugated with asparagine (CHOS-N) and glutamine (CHOS-Q) to inhibit HIV-1 was explored. CHOS-N and CHOS-Q could protect human T cells from HIV-1 infection and cell death (Table 3) [133].

### 4.6. Antimicrobial Activity

Due to the enhanced antimicrobial resistance in various microbes, there is an urge for new compounds with bacterial inhibition properties. CHOS also possesses antimicrobial activities (Table 2), which have gained a lot of attention in recent years. The antibacterial potential of CHOS is typically regulated by several mechanisms, including (i) intracellular constituent leakage caused by bacterial cell wall rupture [134,135], (ii) the destruction of mRNA and protein translation machinery and the inhibition of the enzyme DNA gyrase, and (iii) the chelation of essential nutrients, metals, and spore elements [135]. The MW, DP, and chemical modifications of a CHOS mostly affect its antimicrobial activity. In comparison between CHOS with medium and high MW, CHOS with lower MW often showed higher antibacterial activities. Gram-positive and Gram-negative bacteria were more inhibited by CHOS with a MW of 0.7 kDa and a DP of 2–8 than CHOS with a MW of 1.2 kDa and a DP of 3–13 [30]. The antibacterial activities of CHOS with varying MWs against *Vibrio vulnificus* were compared, in which CHOS possessing a MW of 1 kDa showed higher inhibition than larger-MW CHOS (10 kDa) [136]. Fernandes et al. [137] found that CHOS (<3 and <5 kDa) had higher inhibitory effects against *Klebsiella pneumoniae*, *Escherichia coli*, and *Pseudomonas aeruginosa* than CS of various MWs (628, 591, and 107 kDa). CHOS (0.5 and 1%) also showed antifungal properties by inhibiting the growth of the dermatophytic fungus *Trichophyton rubrum* [138]. CHOS also inhibited the growth of different aflatoxin producers, namely *Aspergillus flavus* and *A. fumigatus*, via the deformation of mycelia and the induction of the leakage of cytoplasmic contents [139].

Multiple techniques have been used to improve the bacterial inhibitory effects of CHOS. Due to an increase in the quaternary amine’s alkyl chain length, *N*-quaternization increased the antibacterial activities of CHOS [140,141]. Generally, quaternary ammonium salt augments the electro-positivity and water solubility of CHOS [142]. Thus, highly electropositive CHOS might more strongly adhere to negatively charged bacterial surfaces. Additionally, the antibacterial capabilities were successfully improved by conjugating different PPNs with CHOS [64]. Squid pen CHOS conjugated with EGCG more effectively inhibited both pathogenic and foodborne bacterial strains as indicated by the lowest minimum inhibitory and bactericidal concentrations (MIC and MBC, respectively) compared to native squid pen CHOS [69]. In another study, a CHOS-catechin conjugate had the lowest minimum MIC and MBC values of 0.256 and 1.024 mg/mL, respectively, against *Vibrio parahaemolyticus* [143]. The free protonated amino group (NH_2_^+^) of a CHOS-catechin conjugate might interact with the negative charge of lipopolysaccharide, which could cause a disruption of the cell membrane and induce the leakage of intracellular constituents such as potassium ion (Figure 3A). In addition, a disruption of the cell membrane is also indicated by an increase in the extracellular malondialdehyde content when treated with a CHOS-catechin conjugate. Furthermore, a shucked Asian green mussel inoculated with 10^6^ CFU/g of *V. parahaemolyticus* was treated with a CHOS-CAT conjugate at 0.512 mg/mL. As a result, the CHOS-CAT conjugate was able to inactivate *V. parahaemolyticus* in the mussel meat by about 84% when incubated at 37 °C for 6 h. Another CHOS-PPN conjugate, namely CHOS-EGCG conjugate, was able to inhibit the growth of *Listeria monocytogenes* at a higher extent when compared with a native CHOS and EGCG alone [144]. The penetration of the CHOS-EGCG conjugate into the bacterial cell via the disruption of the cell membrane resulted in the leakage of protein (Figure 3B). The compound could bind to the negative charges of phosphate groups in the DNA structure. Moreover, CHOS-gallic acid conjugate has been exploited for its antifungal property against *A. versicolor* F1/10M9, *A. montevidensis* F1/30M20, and *P. citrinum* F1/23M14 isolated from dried salted fish (Table 3) [139,145]. MIC and minimum fungicidal concentration (MFC) values were in the ranges of 0.625–2.5 mg/mL and 1.25–10 mg/mL, respectively, against the aforementioned fungal strains. The ergosterol biosynthesis enzymes are suppressed by gallic acid, which could cause the loss of fungal cell membrane integrity and, subsequently, cell death (Figure 3C). Moreover, CHOS-GAL conjugate at the concentration of 5 mg/mL could inhibit the spore germination of *A. versicolor* and *P. citrinum*.

**Table 2 foods-12-03854-t002:** Bioactivities and applications of chitooligosaccharides.

Sources	Applications and Bioactivities	References
Shrimp shell CHOSs	1. Antioxidant activities 2. Antimicrobial activities	[30]
	Inhibition of α-amylase and α-glucosidase activities	[64,98]
	Decrease in TNF-α, NO, and IL-6 levels in LPS-induced RAW 264.7 macrophages Reduce reactive oxygen species formation, NF-κB upregulation, phosphorylation of Erk1/2 and Akt, and Nrf2/HO-1	[109]
	Enhance shrimp oil emulsion stability with augmented oxidative stability	[91]
Squid pen CHOSs	1. Antioxidant activities 2. Inhibit lipid oxidation and extend shelf life of refrigerated sardine surimi gel 3. Antimicrobial activity against *Pseudomonas aeruginosa* PSU.SCB.16S.11, *Listeria monocytogenes* F2365, *Vibrio parahaemolyticus* PSU.SCB.16S.14, *Staphylococcus aureus* DMST 4745, and *Salmonella enterica* serovar Enteritidis S5–371	[9]
	Inhibit discoloration and shelf-life extension of yellowfin tuna slices at 4 °C in combination with oxygen-based MAP	[88]
	Inhibit discoloration and shelf-life extension of yellowfin tuna slices at 4 °C	[89]
	Extend shelf-life extension of Asian sea bass slices stored at 4 °C for 12 days in combination with high-voltage cold atmospheric plasma	[90]
	Enhance gel strength of surimi gel	[146]
CHOSs	Enhance DPPH radical scavenging activity, metal chelating activity, and reducing power	[84]
	Lower blood glucose levels in diabetic rats	[18]
	1. Improve glucose metabolism via lowering fasting blood glucose and insulin levels 2. Upregulation in mRNA expression of GLUT 4 in muscle and adipose tissue	[100]
	Reduce activity of carbohydrase enzymes	[101]
	Reduce blood glucose and lipid levels via downregulations of PEPCK, FBPase, and G6Pase	[104]
	Reduce TNF-α, IL-8, COX-2, and PGE2 levels in LPS-induced A549 cells	[65]
	Suppress PC3 (prostate cancer cell), A549 (lung cancer cell), and HepG2 (hepatoma cell) growths	[116]
	Inhibit SARS-CoV-2 through suppression of both RdRp and E genes	[131]
	Antimicrobial activity: inhibition of *Vibrio vulnificus*	[136]
	Antimicrobial activity: inhibition of *Klebsiella pneumoniae*, *Escherichia coli*, and *Pseudomonas aeruginosa*	[137]
	Alleviated TNF-α production	[110]

### 4.7. Other Applications of CHOS and Its Derivatives

In addition to several applications of CHOS, it attenuated oligomeric Aβ1-42-induced neurotoxicity via the repression of oxidative stress and blocked the Aβ-mediated phosphorylation of c-Jun N-terminal kinase in rat hippocampal neurons by inhibiting fibril formation and disrupting preformed fibrils in a dose-dependent manner [146]. The cell viability of neurons was about 91% when treated with both 5 μM of Aβ42 and 0.5 mg/mL of COS for 48 h. The cell viability of neurons decreased to 76% when exposed to 5 μM Aβ42 alone for 48 h [146]. Moreover, the encapsulated form of the CHOS showed enhanced bioavailability and alleviated liver fibrosis in mice by inhibiting the apoptosis of liver cells and angiogenesis [147].

CHOS also showed improved gelling properties of sardine surimi gel (Table 2), in which the breaking force was increased with the addition of a squid pen CHOS (10 g/kg) [148]. The increased breaking force was more likely associated with the cross-linking ability of the CHOS, in which it might act as the acyl acceptor for the transglutaminase during the setting phenomenon. In addition, the CHOS was also used as a thickener in the Pickering emulsion as prepared in the form of glycated whey protein isolate (gWPI)–CHOS nanoparticles [149]. When compared to emulsions made with WPI, gWPI, or WPI-CHOS alone, those produced nanoparticles successfully enhanced the viscoelasticity and stability of the curcumin-loaded Pickering emulsions [149]. CHOS was also incorporated into the nanofiber prepared via the electrospinning of gelatin/CS nanofibers on polylactic acid film. CHOS improved the film’s properties and showed high antioxidant activities, especially at higher levels of incorporated CHOS [150]. Additionally, the films showed antibacterial effects on both Gram-positive and Gram-negative microorganisms [150]. The prepared film could be used for the packaging of several foods, especially fish fillets, which easily spoil. Recently, Chen et al. [151] prepared gelatin/CHOS nanoparticles produced by the Maillard reaction loaded with apple polyphenols, which could be used as good carriers with good antioxidant activities and stability. Various CHOS-PPN conjugates have been explored for their potential toward the inhibition of polyphenol oxidase (PPO) from shrimp cephalothorax [152]. Among all tested CHOS-PPN conjugates, the CHOS-catechin had the lowest IC_50_ (0.32 mg/mL) and inhibited PPO via mixed-type inhibition kinetics. Via molecular docking, the CHOS-catechin interacted with the Tyr208 or Tyr209 in the active site of the PPO through hydrogen bonds, van der Waals interactions, and hydrophobic interactions. As a result, it could function as a melanosis-preventative agent in shrimp and other crustaceans.

**Table 3 foods-12-03854-t003:** Bioactivities and applications of chitooligosaccharide (CHOS) derivatives.

CHOS Derivatives	Bioactivities and Applications	References
CHOS-caffeic acid	Antioxidant activities	[67]
CHOS-phenolic acid salts	Antioxidant activities	[68]
CHOS-gallic acid	1. Antioxidant activities 2. Antimicrobial activities	[69]
	Antifungal property	[139,145]
	Reduce TNF-α, IL-8, COX-2, and PGE2 levels in LPS-induced A549 cells	[65]
	Inhibit SW620 colon cancer cell growth via upregulation of p53, p21, Bax, caspase 9, and caspase 3 proteins	[128]
CHOS-caffeic acid	Enhance antioxidant property of film	[75]
Sulfated-CHOS	1. Protect pancreatic β-cells and MIN6 cells from H_2_O_2_-induced apoptosis	[105]
	2. Downregulation of H_2_O_2_-induced Bax, Caspase-3, and NF-κB/p65 activation	
	3. Upregulation of Bcl-2	
	Inhibit influenza A virus effects in MDCK cells	[60]
	Inhibit HIV-1-induced syncytia formation, lytic effect, and p24 antigen production	[132]
CHOS-biguanide	1. Lower the blood glucose level by protecting the insulin signaling system 2. Delay β-cell apoptosis 3. Improve β-cell function and promote insulin secretion	[107]
4-hydroxybenzyl-CHOS	Suppress iNOS, COX-2, and MAPK activation and NF-κβ and Iκβα degradation	[63]
Apple polyphenols-CHOS microcapsule	Increase IL-10 level Decrease NO and TNF-α levels	[113]
AE-CHOS and DEAE-CHOS	Inhibit AGS gastric cancer cell growth via regulation of p53, p21, Bcl-2, and Bax proteins	[126]
Carboxymethyl-CHOS	Suppress tumor growth in H22 tumor-bearing mice model	[130]
CHOS-asparagine and CHOS-glutamine	Protect human T cells from HIV-1 infection	[133]
CHOS-epigallocatechin gallate	1. Antimicrobial activity against *Listeria monocytogenes* and *Escherichia coli* 2. Antioxidant activities	[69,144]
CHOS-catechin	Antimicrobial activity against *Vibrio parahaemolyticus*	[143]
	Inhibit polyphenol oxidase activity	[150]

## 5. Conclusions and Future Prospects

Chitooligosaccharide (CHOS) was prepared through the hydrolysis of chitosan (CS) via physical, chemical, or enzymatic methods. Among all hydrolysis methods, the enzymatic hydrolysis assisted with non-specific enzymes is effective; however, redox pair (ascorbic acid and H_2_O_2_) reaction is gaining interest for hydrolysis, especially high-molecular-weight CS. Its multifaceted benefits such as antioxidant and antimicrobial effects, etc., make it a versatile candidate for improving food quality. Moreover, CHOS with several bioactivities, e.g., anti-inflammatory, anti-cancer, and antidiabetic bioactivities, can play pivotal roles in human health benefits. Furthermore, CHOS can be modified by the introduction of various inorganic and organic functional groups at the C-2, C-3, or C-6 positions on the pyranose ring using different methods (carboxylation, quaternization, sulfation, conjugation with polyphenols, etc.), in which enhanced aforesaid bioactivities are achieved. Therefore, CHOS and their various derivatives have become promising alternatives in the fields of food and medicine. They can serve as excellent candidates for developing active packaging or coatings due to their antimicrobial properties that can extend the shelf life of perishable products. They also play a prominent role in the development of functional foods with improved health benefits or disease therapies due to anti-inflammatory, anti-cancer, antidiabetic, and other bioactivities. In addition, the production of CHOS could significantly reduce waste losses, especially from the seafood industry, and ensure food security. However, the applications of CHOS and their derivatives in food and medicine have several challenges and limitations including the high production cost of high-purity CHOS and their derivatives. Therefore, the development of scalable and cost-effective production methods is essential for their wider uses. Also, CHOS and their derivatives may encounter regulatory hurdles in some countries, particularly in terms of safety and labeling, which can hinder their adoption in food and pharmaceutical applications. Therefore, comprehensive research and clinical trials are needed to establish their safety and efficacy for their utilization as a functional ingredient or nutraceutical. Future research to overcome existing hurdles is crucial for the potential uses of CHOS and their derivatives.

## Figures and Tables

**Figure 1 foods-12-03854-f001:**
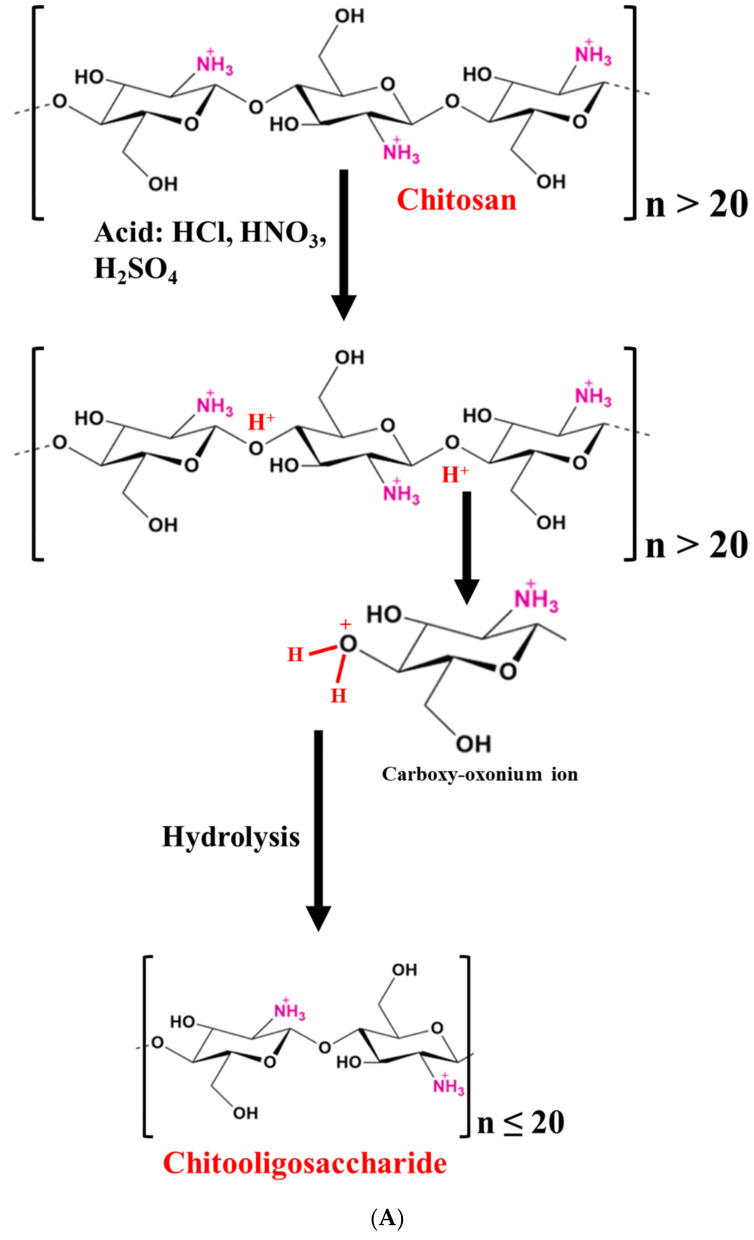
(**A**) Flowchart of chitooligosaccharide production via acid hydrolysis of chitosan. (**B**) Flowchart of chitooligosaccharide production via hydrolysis of chitosan using redox pair reaction.

**Figure 2 foods-12-03854-f002:**
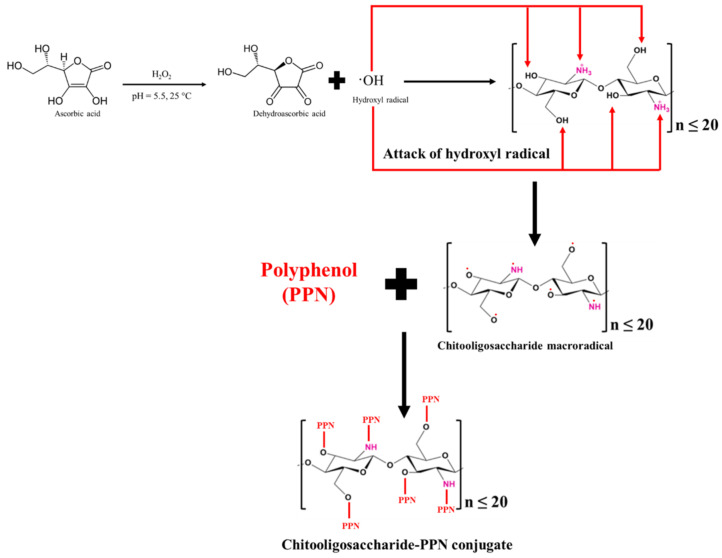
Flowchart for the formation of chitooligosaccharide-polyphenol conjugate using free radical grafting method.

**Figure 3 foods-12-03854-f003:**
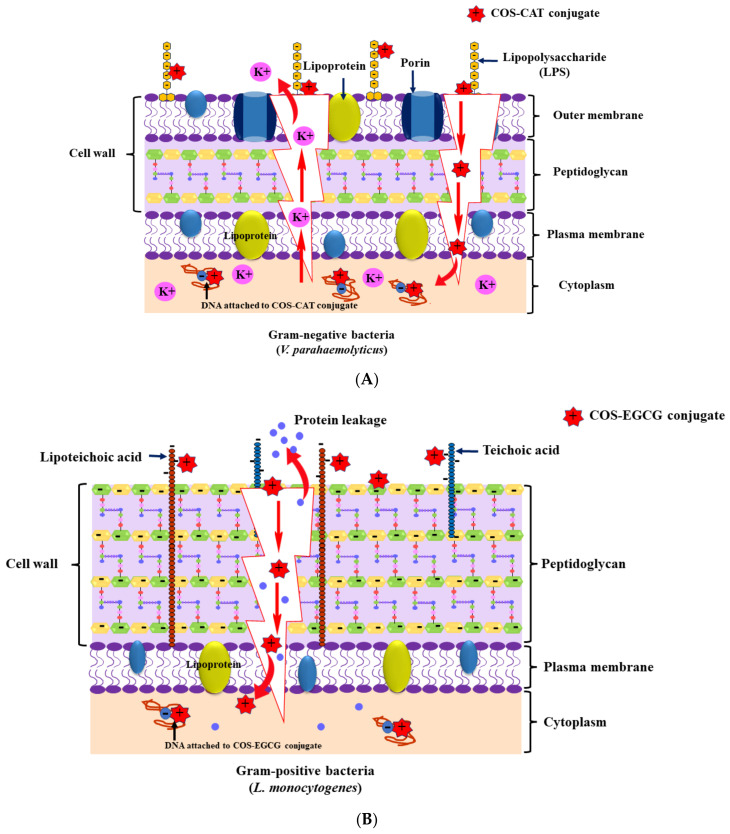
(**A**) Inhibition mechanism of chitooligosaccharide-catechin conjugate against *Vibrio parahaemolyticus*. (**B**) Inhibition mechanism of chitooligosaccharide-catechin conjugate against *Listeria monocytogenes*. (**C**) Inhibition mechanism of chitooligosaccharide-catechin conjugate toward fungal cell.

## Data Availability

The data that support this study are available from the corresponding authors upon request.

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
