# Peer review of "Chitooligosaccharide and Its Derivatives: Potential Candidates as Food Additives and Bioactive Components"

_foods, 2023, doi:10.3390/foods12203854_

Round 1

Reviewer 1 Report

The manuscript has been well written and quite interesting to the relevant fields. I just suggest increasing the Figure quality (>300 dpi to 600 dpi) and words in the Figures.

The review provides an overview of Chitooligosaccharide (CHOS) and its derivatives, highlighting their preparation methods and potential applications in both the food and medical fields. The study effectively summarizes the key points of the review, providing a clear understanding of the topic and its relevance in both food and medical fields.  

It is interesting to the relevant areas.  

The core concepts of CHOS preparation and its potential applications in the food industry are not entirely novel, the specific methods, modifications, and potential solutions to commercialization challenges discussed in this research may contribute to the originality of the work.   

The paper has been written well and the text is clear and easy to read.  

The conclusions are consistent with the evidence and arguments presented.

They do address the main question posed.  

well prepared.

Author Response

Comments and Suggestions for Authors

The manuscript has been well written and quite interesting to the relevant fields. I just suggest increasing the Figure quality (>300 dpi to 600 dpi) and words in the Figures.

******Thank you for the valuable suggestion. The quality of both figures has been improved and font size has been enlarged for better visualization. Please see the edited figures.

The review provides an overview of Chitooligosaccharide (CHOS) and its derivatives, highlighting their preparation methods and potential applications in both the food and medical fields. The study effectively summarizes the key points of the review, providing a clear understanding of the topic and its relevance in both food and medical fields. 

It is interesting to the relevant areas. 

The core concepts of CHOS preparation and its potential applications in the food industry are not entirely novel, the specific methods, modifications, and potential solutions to commercialization challenges discussed in this research may contribute to the originality of the work.  

The paper has been written well and the text is clear and easy to read. 

The conclusions are consistent with the evidence and arguments presented.

They do address the main question posed.

******Thank you for your invaluable comment. To provide some backgrounds to the readers, the general concepts related with CHOS and its uses in food industry were included in the text. Your understanding of our work is highly appreciated.

Reviewer 2 Report

This manuscript is organized well. However, a few major/minor corrections are needed before consideration.

Line 29/ 30 sentences should be one sentence in the better way. At present, there is a lack of reading flow. “Crustacean processing wastes are generated at 6-8 metric tonnes (MT) across the globe annually. Approximately 1.5 MT were reported in southeast Asia [1]”.

In the title, the first letter of each word should be capitalized. What is “functional ingredient” need to change it in the better terminology.

Line 43. Superscript of Mw? Check the whole manuscript.

Introduction don’t have enough background information related to this review manuscript. Need to improve better way by adding recent references.

Figure 1 is not proposed mechanism its already reported previously so authors need to cite previous work.

Please describe the effect of CHOS applications with respect to molecular weight variation and its DP.

Authors mention CHOS are within 20kDa further authors’ mention CHOS are in between 20-50 kDa comment about it (Lie 514 and other places). Check the accuracy statement and correct it in the manuscript.

All the figures should be properly cited if the authors do not create it.

Insert the following suggested manuscript to strengthen the text of the manuscript Polymers 15 (1) (2023) 132; Int. J. Mol. Sci. 201516(5), 10526-10536 International journal of biological macromolecules 136 (2019) 661-667; Int. J. Mol. Sci. 201819(8), 2197; and https://link.springer.com/chapter/10.1007/978-3-030-92806-3_4.

Figure 3 A and Fig 3B resolution is poor. Please improve it.

Global research trends information is missing regarding CHOS? Please include it.

In conclusion, there is almost no information.

Future perspectives should be discussed critically.

Recent references are very few as compared to old ref. Preferable within five years references.

English is fine. Check typos errors.

Author Response

Comments and Suggestions for Authors

This manuscript is organized well. However, a few major/minor corrections are needed before consideration.

******Thank you for your invaluable comment. Your understanding of our work is highly appreciated. The manuscript has been improved in clarity and quality as per the reviewer’s comments and suggestions. All queries have been responded and the corrections have been made as highlighted in yellow color.

Line 29/ 30 sentences should be one sentence in the better way. At present, there is a lack of reading flow. “Crustacean processing wastes are generated at 6-8 metric tonnes (MT) across the globe annually. Approximately 1.5 MT were reported in southeast Asia [1]”.

******Thank you for your comment. The given sentence has been rewritten for better understanding. Please see line 29-30.

In the title, the first letter of each word should be capitalized. What is “functional ingredient” need to change it in the better terminology.

******Thank you for your comment. The first word of each word in the title has been capitalized. Please see the title of the manuscript.

******‘Functional ingredient’ is a bioactive compound that can be used in the manufacture of functional food products. It can be obtained from several sources including seafood processing leftover. It has been used particularly in the food and dietary supplement industries. Nevertheless, the authors have changed ‘Functional ingredient’ to ‘Bioactive component’ to be more specific terminology and better understanding.

Line 43. Superscript of Mw? Check the whole manuscript.

******Authors rechecked the manuscript carefully and found that molecular weight was abbreviated as ‘MW’ throughout the manuscript. Also, changes have been made where typological errors are found.

Introduction don’t have enough background information related to this review manuscript. Need to improve better way by adding recent references.

******Thank you for your insightful comment. The introduction given in this review manuscript broadly covers the background of six major areas including waste generation and environmental concerns, bioactive compounds in shrimp processing waste, chitosan and chitooligosaccharide (CHOS) extraction, diverse applications of CHOS, enhancing CHOS bioactivities, and utilization of CHOS and its derivatives in food and nutraceuticals. However, authors also included additional information, especially the updated information, for quality improvement of the review article. Please see line 43-46 and 51-52. Moreover, recent citations within five years have been added for updated information while mostly out-of-date references have been removed from text.

Figure 1 is not proposed mechanism its already reported previously so authors need to cite previous work.

******Thank you for your comment. Authors agreed that the mechanism shown in Figure 1 has been already reported in various published manuscripts but in the text format. The lack of pictorial representation of those mechanisms intrigued authors to create it in an interactive form. This figure can provide a better understanding for readers. As a consequence, the authors solely designed Figure 1 based on the existing knowledge in the various published literatures rather than opting for one study. Therefore, the authors are not able to add all references in the figure caption. Also, a discussion related to Figure 1 had already been included in the text with proper citations. In addition, the authors edited the figure caption to avoid confusion. Please see the caption of Figure 1.

Please describe the effect of CHOS applications with respect to molecular weight variation and its DP.

******Thank you for your comment. So far, CHOS from different sources has been prepared using different methods. Some CHOS has been commercialized. It is true that bioactivities of CHOS e.g., anticancer or antitumor can be varied in different testing systems e.g., in vitro and in vivo systems, despite having the same molecular weight (MW) or DP. Moreover, the bioactivities of CHOS are also affected by the preparation method and arrangement of glucosamine and N-acetyl glucosamine units in the CHOS chain. Therefore, it is nearly impossible to define the MW or DP of CHOS with a certain range for a single application.  Nevertheless, CHOS having low MW, which also corresponded with low DP, generally showed higher bioactivities in different systems than their high MW counterparts. Such an information was given in the manuscript at different places. Please read the manuscript carefully.

Authors mention CHOS are within 20kDa further authors’ mention CHOS are in between 20-50 kDa comment about it (Lie 514 and other places). Check the accuracy statement and correct it in the manuscript.

******Thank you for the insightful comment. The authors rechecked the information about the MW of CHOS and generally, it was less than 3.9 kDa. With some exceptions in the existing literature, MW of CHOS was reported higher than the aforementioned value. This might be due to application of ineffective hydrolysis method. The authors included one phrase regarding the MW of CHOS in the introduction section to avoid confusion for readers. Please see line 45-46. Moreover, the authors carefully rechecked the manuscript and confirmed that information about MW of CHOS was solely based on the existing literatures.

All the figures should be properly cited if the authors do not create it.

******Thank you for your comment. The figures used in this manuscript are solely prepared by authors based on the existing knowledge cited in the text. Authors realize well to avoid the plagiarism.

Insert the following suggested manuscript to strengthen the text of the manuscript Polymers 15 (1) (2023) 132; Int. J. Mol. Sci. 2015, 16(5), 10526-10536 International journal of biological macromolecules 136 (2019) 661-667; Int. J. Mol. Sci. 2018, 19(8), 2197; and https://link.springer.com/chapter/10.1007/978-3-030-92806-3_4.

*******Thank you for your invaluable comment and suggested manuscript. The current review article focuses on the preparation of chitooligosaccharide and its derivatives along with their potential bioactivities and applications. The following references ‘Polymers 15 (1) (2023) 132’ and ‘International journal of biological macromolecules 136 (2019) 661-667’ are focused on the application of chitosan and modification of chitosan, which are not aligned with the scope of the current manuscript. Therefore, authors cannot include these references in the present manuscript. Apologies for it.

******Nevertheless, the authors gathered relatable information and added it to the text for further explanation from the following references ‘Int. J. Mol. Sci. 2015, 16(5), 10526-10536’, ‘Int. J. Mol. Sci. 2018, 19(8), 2197’, ‘https://link.springer.com/chapter/10.1007/978-3-030-92806-3_4’. The aforementioned references are in the scope of this manuscript. Please see line 49, 55, and 787-793. Thank you for your invaluable suggestion to strengthen the quality and merit of manuscript.

Figure 3 A and Fig 3 B resolution is poor. Please improve it.

******Thank you for your valuable comment. Figure 3A and 3B has been provided with enhanced resolution. Please see the edited figures.

Global research trends information is missing regarding CHOS? Please include it.

******Thank you for your useful comment. Worldwide research trends about CHOS could give more insightful information, however detailed information such as article published by number, year, country, subject area, etc. are not in the scope of this review manuscript. The current manuscript focuses on the contents of CHOS and its derivatives along with their several bioactivities. Brief information was provided in the introduction to highlight research trends. Please see the last paragraph of introduction (line 53-66), which included the research trend of CHOS covering the detailed information about the enhanced bioactivities via several methods and applications of CHOS and CHOS derivatives in different fields for the full exploitation of CHOS.

In conclusion, there is almost no information.

******Thank you for your comment. The conclusion has been extensively edited for a better understanding of the whole content. Please see Section 5.

Future perspectives should be discussed critically.

******Thank you for the valuable comment. The reviewer’s suggestion is taken into consideration. Future perspectives of CHOS and its derivatives have been discussed precariously. In addition, prospective challenges or hurdles for the application of CHOS and its derivatives have been added. Please see section 5.

Recent references are very few as compared to old ref. Preferable within five years references.

******For a better understanding of readers, there is a necessity to provide basic principles along with important mechanisms in the text, which were comprehensively discussed with the support from the previous literatures. Therefore, the author inevitably cited some old references. However, as per the suggestion of reviewers, the latest references have been added to the text to gain the updated information. Please see the highlighted references in the bibliography.

Reviewer 3 Report

Detailed comments are included in the review file attached below.

The language level is generally good. The manuscript contains some minor linguistic errors, which can be improved.

Author Response

Comments and Suggestions for Authors

A brief summary:

The aim of this review article was to describe the preparation of chitooligosaccharide (CHOS) and its derivatives and explore their applications in food as additives or nutraceuticals. Additionally, recent progress in translational research and in vivo studies related to CHOS and its derivatives in the medical field was presented.

General comments:

In the manuscript, in addition to in-depth characterization of the methods of chitooligosaccharide (CHOS) preparation and its modification in order to improve their bioactivities, also applications of CHOS and CHOS derivatives in food as well as its nutraceutical aspect were also presented.

On the basis of the presented data it can be concluded that the pharmaceutical industry has taken considerable interest in the anti-inflammatory, anti-obesity, neuroprotective, and anticancer properties of CHOS derivatives and also in the food industry CHOS and CHOS derivatives can be exploited for food applications.

Analysing overall merit, the work provides a piece of useful information towards the current knowledge in this field. This work is a comprehensive review and consist a scientific and practical support for the application of CHOS and CHOS derivatives in the food but also in medicinal purposes.

Rating interest to the readers and taking all the information from this review article into account, in the reviewer’s opinion this work can certainly interest the readers, especially due to their willingness to ensure their health and safety by consuming safe foodstuffs. When it comes to the presentation of information, it is a broad manuscript, comprehensive in terms of topics, and at the same time it contains a lot of useful and interesting information for potential readers. In the reviewer’s opinion the manuscript’s topic is well presented, however has some imperfections reducing the quality of work and undoubtedly should be eliminated.

References are well-developed and exhaustive. Furthermore, the selection of references is good, but observed excessive self-citation should be reduced.

******Thank you for your invaluable comment. Your understanding of our work is highly appreciated. The manuscript has been improved in clarity and quality as per the reviewer’s comments and suggestions. All queries have been responded and the corrections have been made as highlighted in green color. For the self-citation, authors have conducted the research on CHOS and its derivative intensively. Thus, authors have the sufficient information for the comprehensive review along with the information or papers from other research groups.

The specific comments to the manuscript are as follows:

  • Lines 75-77 - The table 1 is referenced in paragraph 2.1.1. Acid hydrolysis, and this method is not included in the table at all. Authors wrote: ‘Several investigations have been conducted to better understand the process of CS hydrolysis via acids (Table 1) [23-26]’, whereas in the table 1 these data were not presented at all.

******Thank you for your comment and sorry for the mistake. The studies related to acid hydrolysis have been included in Table 1. Please see Table 1.

  • Line 100 - the figure caption format is inappropriate, the name: Figure 1 was not bolded and a semicolon was used instead of a dot;

******Sorry for mistake. The figure caption has been edited. Moreover, ‘Figure 1’ has been bolded and the semicolon was replaced with a dot as suggested by the reviewer. Please see line 105.

  • Line 103 - the name of the Table 1 should contain information that data presented in this table is selected, and does not present all the methods, which was described in the text of the manuscript.

******Thank you for your comment. The main objective of Table 1 is to provide the preparation details of chitooligosaccharide (CHOS) using various methods along with analytical techniques used for the characterization and characteristics of CHOS. The authors believed that supplementary information provided in the form of method names in Table 1 would be relevant for readers who are seeking in-depth information. Authors avoid repeating the presentation of same information in both text and table.

  • Analogical problem in the subsection 2.1.2. Oxidative hydrolysis (as in the subsection 2.1.1.) – Lines 109-110 - Authors wrote: ‘Several reports have been available on the hydrolysis of CS with the aid of H2O2 (Table 1) [28,30-32]’ , whereas in this Table 1 data from different references about usage of hydrogen peroxide were presented and cited, so reference no. 30,31,35, while Authors mentioned no. 28 and 30-32.

******Thank you for your valuable comment. The authors carefully rechecked the text for the citations. Out of references highlighted by the reviewer, the reference number 28 has been removed from the aforesaid lines to avoid confusion, while the reference number 32 has been included in Table 1 for better understanding of readers.

  • Analogical problem in the subsection 2.2. Physical methods, Authors wrote (Lines 136-137): ‘CHOS has been created using physical methods such as microwave, lambda radiation, and high-intensity ultrasonication (Table 1) [34] and in the Table 1 there is no such a reference cited.

******Thank you for your invaluable comment. The authors carefully checked the citations throughout the text. Reference number 34 cited in aforesaid lines is a review article that supports general information. Therefore, it cannot be included in Table 1, which possesses the references related to research articles. To avoid confusion among readers, the authors have quoted Table 1 in line 145-146.

  • Similar problem in the subsection 2.3. Enzymatic hydrolysis, Authors wrote (lines 151- 152): “ As a result, non-specific enzymes including lipase, carbohydrase, proteases, etc. have been employed for CS hydrolysis (Table 1) [9,20,43]’ , whereas in the Table 1 data from different references (no. 41,44, 45) were cited but reference no. 20 and 43 were not cited at all. Therefore, the reader gets the impression of a mess when quoting and presenting the quoted data in the table 1 and the text of the manuscript.

******Thank you for your comment. Similar to the previous query, the authors would like to explain that aforesaid line is a general statement, which was supported by reference number 9, 20, and 43. Among those, reference number 20 and 43 are review articles while reference number 9 is a research article. On the other hand, the information provided in Table 1 was based on the research articles. Therefore, the authors cannot include reference numbers 20 and 43 in Table 1. Nevertheless, the reference number 9 has been removed from lines mentioned by the reviewer to avoid confusion. Moreover, Table 1 has been quoted in line 162-163 for better clarity and understanding of readers.

  • Line 134 - the figure caption format is also inappropriate, the name: Figure 2 was not bolded and a semicolon was used instead of a dot;

******Thank you for your comment. The figure caption has been edited. Please see line 367. Moreover, ‘Figure 2’ has been bolded and the semicolon has been replaced with a dot as suggested by the reviewer. Similarly, other figure captions have been edited according to the reviewer’s suggestion. Please see the edited figure caption of each figure.

  • line 246 – ‘two h’ instead of 2 h;

******Authors changed ‘two h’ to ‘2 h’. Please see line 247.

  • Unnecessary free lines from lines 528 to 535;

******Thank you for your comment. This review manuscript focuses on the bioactivities of chitooligosaccharide (CHOS) and its derivatives, in which the antioxidant potential of various CHOS-polyphenol conjugates in different systems was reported. The information was mentioned between L528 and L535 (now line 531-538). Therefore, the discussion or information given in the aforementioned lines provided by the authors were well aligned with the scope and objective of the current review manuscript.

******Moreover, the authors would like to explain that a review article is a type of academic paper that summarizes and evaluates existing research on a particular topic or field. The work explained in lines mentioned by the reviewer has been already published, and its inclusion is necessary for a better understanding of content. Therefore, the authors would like to keep the information in the manuscript that provided in those lines.

  • A dot is missing at the end of each table name.

******The full stop has been added at the end of all table names.

  • Unnecessary free lines from lines 560 to 575;

******Thank you for your comment. Section 4.2. describes the antidiabetic potential of the chitooligosaccharide and its derivatives, in which line 560-575 describes the role of digestive enzyme (α-amylase and α-glucosidase) in increasing blood sugar levels, their effective inhibition by CHOS and its mode of inhibition. All the information provided in those lines was necessary to provide the background information related to the study. It would also enhance the transparency and clarity of the article by providing in-depth information, which further helps readers to gain the better understanding of the context of research and its progression. Therefore, this in-depth information is critical for strengthening the manuscript in a meaningful way.

  • Italics is intended for the names of microorganisms, although it was not used for their names in the tables (2 and 3).

******The names of microorganisms have been italicized in both Tables 2 and 3.

  • in the whole manuscript, when surnames are cited in the text, the formulation ‘et al.’ was written unnecessary in italics and there is no need for a comma before et al. (e.g. line 764); The ‘Abstract’ and ‘Conclusions’ should be more different from each other. Authors should consider reformulating them.

******Thank you for your comment. Throughout the manuscript, ‘et al.’ has been changed to regular fonts from italics. Moreover, the comma has been removed from ‘et al.’ as suggested by the reviewer.

******Authors rechecked the abstract and conclusion and both have been edited extensively. Brief information about analytical techniques, the name of CHOS derivatives, and their bioactivities have been added. Moreover, the scope of CHOS derivatives as food additives has been included for better understanding. Similarly, the conclusion of the manuscript has been rewritten more critically. The most important findings of this review article have been highlighted. Moreover, prospects and limitations or research gaps for the application of CHOS and its derivatives have been addressed in the conclusion. Please see the edited abstract and conclusion.  

  • Furthermore, the number of self-citations in this article reaches 18, it is understandable that the authors have extraordinary experience in this field, but it is exceptionally visible both in the text when citing their works and in the list of references.

******Thank you for your comment. The authors advocated that it is important to maintain a balance when citing own work to ensure the overall quality and objectivity of the research. Self-citations aided in acknowledging the author's previous work, providing context for the current research, and demonstrating their expertise in the field. This can provide readers with confidence in the author's credibility.

******Also, the reviewer mentioned about removal of unnecessary lines in their previous comments from the text without providing any convincing reason for it. In those lines, authors cited their work, which was related to the scope of this review article and published in various peer-reviewed journals. It seems that reviewer wants to remove credible work done by authors in the field of chitooligosaccharide.

******Additionally, self-citations should be seen as a positive aspect. When authors cite their work, the readers can easily trace the evolution of ideas, methodologies, and results. Therefore, a high number of self-citations does not indicate bias or a lack of objectivity. Instead, it may reflect the deep expertise of authors and their significant contributions to the field. Thus, the authors strongly advocate for keeping their all citations in the manuscript, which would provide the in-depth information and enhance the transparency of the article for helping the readers to better understand the research context and its progression.

English level:

The language level is generally good. The manuscript contains some minor linguistic errors, which can be improved. Among them, the following can be mentioned e.g.:

  • Line 275-277 – grammar mistake

******Thank you for your comment. The manuscript has been thoroughly checked for grammatical and typological errors and necessary corrections have been made at respective places. Moreover, the manuscript has been proof-read by a native speaker.

In my opinion, the significance of the information presented in the manuscript is high. I have some objections to this article, but after minor revision and some improvements, the manuscript can be published in the ‘Foods’ Journal.